# Unraveling electronic correlations in warm dense quantum plasmas

Tobias Dornheim ●[1,2] ✉, Tilo Döppner ●[3], Panagiotis Tolias ●[4], Maximilian P. Böhme ●[1,2,3,5], Luke B. Fletcher ●[6], Thomas Gawne ●[1,2], Frank R. Graziani ●[3], Dominik Kraus ●[2,7], Michael J. MacDonald ●[3], Zhandos A. Moldabekov ●[1,2], Sebastian Schwalbe ●[1,2], Dirk O. Gericke ●[8] & Jan Vorberger ●[2]

The study of matter at extreme densities and temperatures has emerged as a highly active frontier at the interface of plasma physics, material science and quantum chemistry with relevance for planetary modeling and inertial confinement fusion. A particular feature of such warm dense matter is the complex interplay of Coulomb interactions, quantum effects, and thermal excitations, making its rigorous theoretical description challenging. Here, we demonstrate how ab initio path integral Monte Carlo simulations allow us to unravel this intricate interplay for the example of strongly compressed beryllium, focusing on two X-ray Thomson scattering data sets obtained at the National Ignition Facility. We find excellent agreement between simulation and experiment with a very high level of consistency between independent observations without the need for any empirical input parameters. Our results call into question previously used chemical models, with important implications for the interpretation of scattering experiments and radiation hydrodynamics simulations.

Matter at extreme densities and temperatures displays a complex quantum behavior. A particularly intriguing situation emerges when the interaction, thermal, and Fermi energies are comparable. Understanding such warm dense matter (WDM) requires a holistic description taking into account partial ionization, partial quantum degeneracy, and moderate coupling. Indeed, even familiar concepts such as well-defined electronic bound states and ionization break down in this regime.

Interestingly, such conditions are widespread throughout the universe, naturally occurring in a host of astrophysical objects such as giant planet interiors[1], brown and white dwarfs[2], and, on Earth, meteor impacts[3]. Moreover, WDM plays a key role in cutting-edge technological applications such as the discovery and synthesis of novel materials[4]. An extraordinary achievement has recently been accomplished in the field of inertial confinement fusion at the National Ignition Facility (NIF)[5,6]. In these experiments, both the ablator and the fuel traverse the WDM

regime, making a rigorous understanding of such states paramount to reach the reported burning plasma and net energy gain[7,8].

The pressing need to understand extreme states has driven a large leap in the experimental capabilities; the considerable number of remarkable successes includes the demonstration of diamond formation under planetary conditions[4,9], opacity measurements under solar conditions[10], probing atomic physics at Gigabar pressures[11], and the determination of energy loss of charged particles[12,13]. However, this progress is severely hampered: to diagnose WDM experiments, a thorough understanding of the electronic response is indispensable. Indeed, even the inference of basic parameters such as temperature and density requires rigorous modeling to interpret the probe signal.

Density functional theory combined with classical molecular dynamics for the ions (DFT-MD) has emerged as the de-facto work

---

[1]Center for Advanced Systems Understanding (CASUS), Görlitz, Germany. [2]Helmholtz-Zentrum Dresden-Rossendorf (HZDR), Dresden, Germany. [3]Lawrence Livermore National Laboratory, Livermore, CA, USA. [4]Space and Plasma Physics, Royal Institute of Technology (KTH), Stockholm, Sweden. [5]Technische Universität Dresden, Dresden, Germany. [6]SLAC National Accelerator Laboratory, Menlo Park, CA, USA. [7]Institut für Physik, Universität Rostock, Rostock, Germany. [8]Centre for Fusion, Space and Astrophysics, Department of Physics, University of Warwick, Coventry, UK. ✉e-mail: t.dornheim@hzdr.de

horse for computing WDM properties. While being formally exact[14], the predictive capability of DFT-MD is limited by the unknown exchange–correlation functional, which has to be approximated in practice, and the application of the Born-Oppenheimer approximation. A potentially superior alternative is given by ab initio path integral Monte Carlo (PIMC) simulations[15], which are in principle capable of providing an exact solution for a given quantum many-body problem without any empirical input. Yet, PIMC simulations of quantum degenerate Fermi systems, such as the electrons in WDM, are afflicted with an exponential computational bottleneck, which is known as the fermion sign problem[16,17]. As a consequence, PIMC application to matter under extreme conditions has either been limited to comparably simple systems such as the uniform electron gas model[18], or based on approximations as in the case of restricted PIMC[19–21].

Here, we present a solution to this unsatisfactory situation and demonstrate its capabilities on the example of warm dense beryllium (Be). Since our approach is not based on any nodal restriction, we get access to the full spectral information in the imaginary-time domain[22]. As the capstone of our work, we employ our simulations to re-analyze X-ray Thomson scattering (XRTS) data obtained at the NIF for strongly compressed Be in a backscattering geometry[11]. In addition, we consider a new data set that has been measured at a smaller scattering angle that is more sensitive to electronic correlations. Our unique access to electron correlation functions allows for alternative ways to interpret the XRTS data, resulting in a very high level of consistency. Interestingly, we find a substantially lower density compared to previously used chemical models[11], which has important implications both for our understanding of XRTS measurements and for state-of-the-art radiation hydrodynamics simulations of implosion dynamics.

## Results

### Simulation approach and capabilities

In principle, the PIMC method allows one to obtain an exact solution to the full quantum-many body problem without any empirical input or approximations. However, the application of PIMC to quantum degenerate electrons is severely hampered by the fermion sign problem[16,17]. To circumvent this obstacle, Militzer, Ceperley and others[19–21,23] have successfully employed the fixed-node approximation. This restricted PIMC method allows for simulations of large systems without a sign problem, an advantage that comes at the cost of a de-facto uncontrolled approximation in the form of a trial nodal structure[24,25]. While RPIMC has been successfully applied to a host of materials[21] and experimentally verified, e.g. for CH[2,26], the inherent nodal restriction prevents the usual access of PIMC to the full spectral information in the imaginary-time domain[22,27], preventing a corresponding comparison with XRTS measurements.

In this work, we employ a fundamentally different strategy by carrying out a controlled extrapolation over a continuous variable $\xi \in [-1, 1]$ that is substituted into the canonical partition function[28–30], see the Methods Section. This treatment removes the exponential scaling of the computation time with the system size for substantial parts of the WDM regime without the need for any empirical input such as the nodal surface of the density matrix for restricted PIMC or the exchange–correlation (XC) functional for DFT. At the same time, it retains full access to the spectral information about the system encoded in the imaginary-time density-density correlation function (ITCF), thereby allowing for direct comparison between simulations and XRTS measurements. While this approach had been successfully applied to the uniform electron gas model[29,30], we use it here to study the substantially more complex case of electrons and nuclei in WDM.

In Fig. 1a, b, we show snapshots of all-electron PIMC simulations of $N_{Be} = 25$ Be atoms (i.e., $N_e = 100$ electrons) for the mass density $\rho = 7.5$ g/cc and temperatures $T = 190$ eV and $T = 100$ eV, respectively. The green orbs depict the nuclei, which behave basically as classical point particles, although this is not built into our simulations.

The blue paths represent the quantum degenerate electrons; their extension is proportional to the thermal de Broglie wavelength $\lambda_T = \hbar \sqrt{2\pi / m_e k_B T}$ and serves as a measure for the average extension of a hypothetical single-electron wave function. The interplay of electron delocalization with effects such as Coulomb coupling shapes the physical behavior of the system. In panels c and d, we illustrate PIMC results for the spatially resolved electron density in the external potential of a fixed ion configuration. We find a substantially increased localization around the nuclei for the lower temperature.

Figure 1e, f shows ab initio PIMC results for the full Be system, where both electrons and nuclei are treated dynamically on the same level. Specifically, panel e shows various pair correlation functions, where the red and blue lines correspond to $T = 100$ eV and $T = 190$ eV, respectively. The ion–ion pair correlation function $g_{II}(r)$ [squares] is relatively featureless in both cases. The same holds for the spin-diagonal electron–electron pair correlation function $g_{\uparrow\uparrow}(r) = g_{\downarrow\downarrow}(r)$ [crosses], although the exchange–correlation hole is substantially reduced when compared to $g_{II}(r)$ mainly due to the weaker Coulomb repulsion. In stark contrast, the spin-offdiagonal pair correlation function $g_{\uparrow\downarrow}(r)$ [circles] exhibits a nontrivial behavior and strongly depends on the temperature. While being nearly flat for $T = 190$ eV, $g_{\uparrow\downarrow}(r)$ markedly increases towards $r = 0$ for $T = 100$ eV. This increased contact probability indicates the onset of clustering of two electrons around a single nucleus at the lower temperature, and nicely illustrates the capability of our PIMC simulations to capture the complex interplay of ionization, thermal excitation, and electron–electron correlations. Finally, panel f) shows corresponding results for the ion–ion [squares] and electron-ion [crosses] static structure factor (SSF). These contain important information about the generalized form factor and Rayleigh weight[31], which are key properties in the interpretation of XRTS experiments[11] and a gamut of other applications.

### Application to X-ray Thomson scattering

As a demonstration of the present PIMC capabilities, we re-analyze an XRTS experiment with strongly compressed beryllium at the National Ignition Facility (dataset #3 in ref. 11) and repeated the experiment at a smaller scattering angle to focus more explicitly on electronic correlation effects. It can be assumed that the plasma conditions probed in the two measurements are comparable as (1) the probe times are identical within the timing uncertainty of $\Delta t = 0.05$ ns and (2) the shot-to-shot fluctuations are small and carefully controlled by high-accuracy laser delivery and high reproducibility in target production. Figure 2a shows an illustration of the experimental set-up using the GBar XRTS platform. 184 optical laser beams (not shown) are used for the hohlraum compression[11] of a Be capsule (yellow sphere) which is filled with a core of air. A further 8 laser beams are used to heat a zinc foil generating 9 keV X-rays[32] that are used to probe the system (purple ray). By detecting the scattered intensity (blue ray) at an angle $\theta$, we get insight into the microscopic physics of the sample on a specific length scale; the same microscopic physics can be resolved by our new PIMC simulations, a snapshot of which is depicted inside the Be capsule.

The measured XRTS spectra are shown as the black and red curves in Fig. 2b and have been obtained at scattering angles of $\theta = 120°$ (#3 in ref. 11) and $\theta = 75°$ (new). They are given by a convolution of the dynamic structure factor $S_{ee}(q, \omega)$ with the combined source-and-instrument function $R(\omega)$ [dashed blue]. Since a deconvolution to extract $S_{ee}(q, \omega)$ is unstable, we instead perform a two-sided Laplace transform[22,33,34]

$$F_{ee}(q, \tau) = \mathcal{L}[S_{ee}(q, \omega)] = \int_{-\infty}^{\infty} d\omega \, S_{ee}(q, \omega) e^{-\hbar\omega\tau}; \qquad (1)$$

the well-known convolution theorem then gives us direct access to the imaginary-time correlation function (ITCF) $F_{ee}(q, \tau)$ based on the experimental data, with $\tau \in [0, \beta]$ being the imaginary time and

$\beta = 1/k_B T$ the inverse temperature, see the Methods Section. The ITCF contains the same information as $S_{ee}(q, \omega)$, but in a different representation[27]. A particularly important application of $F_{ee}(q, \tau)$ is the model-free estimation of the temperature[33,34], and we find $T = 155.5 \pm 15$ eV and $T = 190 \pm 20$ eV for $\theta = 120°$ and $\theta = 75°$, respectively.

A second advantage of Eq. (1) is that it facilitates the direct comparison of the experimental observation with our new PIMC results. As a first point, we consider the electronic static structure factor $S_{ee}(q) = F_{ee}(q, 0)$ in Fig. 2c, f for the two relevant temperatures, and the circles and crosses show PIMC results for $N_{Be} = 25$ and $N_{Be} = 10$ beryllium atoms. Evidently, no finite-size effects can be resolved within the given error bars with the possible exception of the smallest $q$ values. A particular strength of the PIMC method is that it allows us to unambiguously resolve the impact of electronic XC-effects. To highlight their importance for the description of warm dense quantum plasmas even in the high-density regime, we compare the PIMC data with the mean-field based random phase approximation[35] (dotted lines). The latter approach systematically underestimates the true $S_{ee}(q)$ and only becomes exact in the single-particle limit of large wave numbers. The blue circles correspond to $S_{ee}(q)$ extracted from the NIF data following the procedure introduced in the recent ref. 36. They are consistent with the PIMC results for $\rho \lesssim 20$ g/cc.

The full ITCF $F_{ee}(q, \tau)$ is shown in panels d and g in the $q$-$\tau$-plane, where the colored surface shows the PIMC results for $\rho = 20$ g/cc, and the dashed blue lines have been obtained from the experimental data via a two-sided Laplace transform, see the Methods Section. We stress that, while the symmetry of the ITCF had been used for the interpretation of XRTS experiments in previous works[33,37,38], the present study compares exact PIMC simulations with an XRTS measurement. Clearly, the ITCF exhibits a rich structure that is mainly characterized by an increasing decay with $\tau$ for larger wave numbers. In fact, this $\tau$-dependence is a direct consequence of the quantum delocalization of the electrons[27,39] and would be absent in a classical system. The NIF data are in very good agreement with our PIMC simulations over the entire $\tau$-range. This can be seen particularly well in panels e and h, where we show the ITCF for the fixed values of $q$ probed in the experiment. We find a more pronounced decay of $F_{ee}(q, \tau)$ with increasing $\tau$ for larger $q$. A second effect is driven by the different temperatures of these separate NIF shots, as a higher temperature leads to a reduction of quantum delocalization and, therefore, a reduced $\tau$-decay. The observed agreement between the PIMC results and the experimental data for different $q$ and temperature is thus nontrivial and constitutes a remarkable level of agreement and consistency between theory and experiment.

In addition, we consider an additional observable that can be directly extracted from the experimental data: the ratio of the elastic to the inelastic contributions to the full scattering intensity $I_{el}/I_{inel}$. In Fig. 3a, the two components are illustrated for the case of $\theta = 75°$. In practice, the elastic signal has the form of the source function [cf. Fig. 2b], and $I_{inel}$ is given by the remainder. The ratio $I_{el}/I_{inel}$ constitutes a distinct measure for the localization of the electrons around the ions on the probed length scale determined by $q$. Therefore, it is highly sensitive to system parameters such as the density, and, additionally, to the heuristic but often useful concept of an effective ionization degree[11]. Yet, the prediction of $I_{el}/I_{inel}$ from ab initio simulations requires detailed knowledge about correlations between all particle species, see the Methods Section. This requisition is beyond the capabilities of standard DFT-MD, but straightforward with PIMC simulations, cf. Fig. 1.

In Fig. 3b, c, we show our simulation results for the $q$-dependence of $I_{el}/I_{inel}$ at $T = 190$ eV and $T = 155.5$ eV, respectively, for three relevant mass densities. These conditions correspond to the XRTS

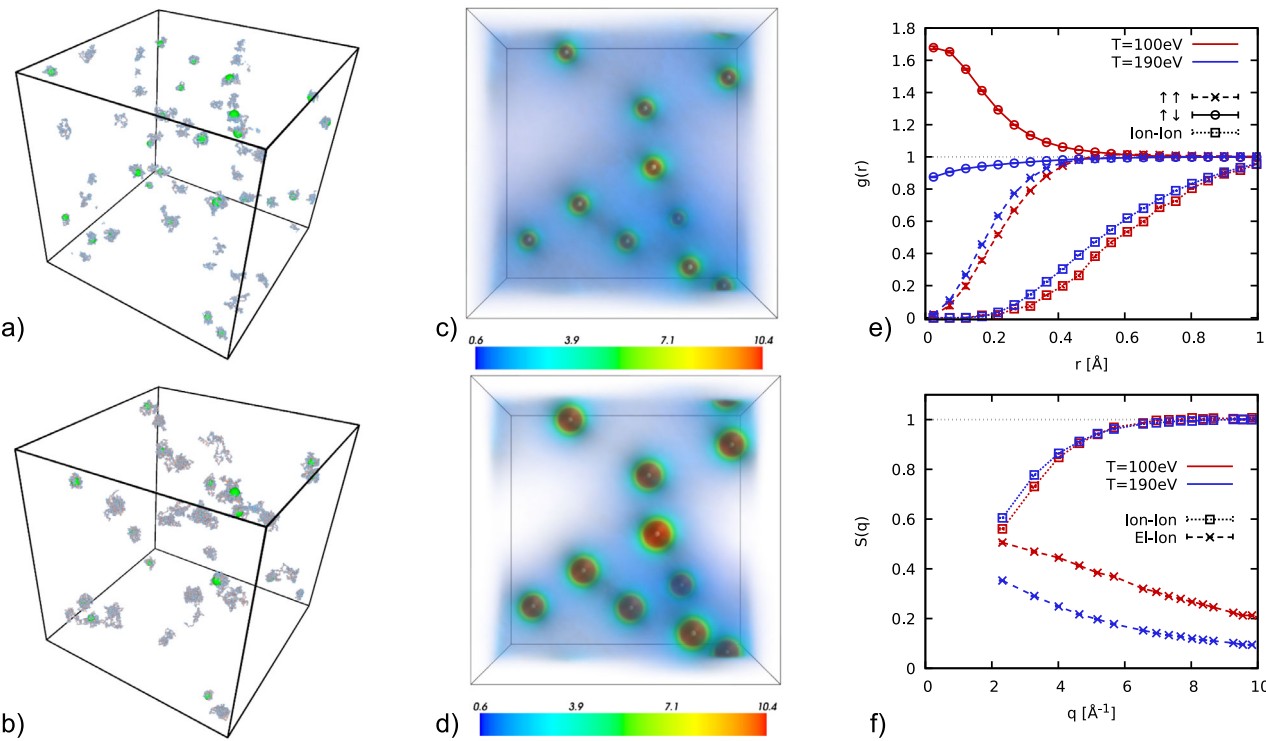

**Fig. 1 | Ab initio PIMC simulations of compressed Be ($\rho = 7.5$ g/cc). a** Snapshot of a PIMC simulation of $N_{Be} = 25$ Be atoms at $T = 190$ eV. **b** Same as (**a**) but for $T = 100$ eV. The green orbs show the ions and the blue-red paths the quantum degenerate electrons; **c, d** electronic density in real space for a fixed ion configuration at the same temperatures, with the colors indicating values in units of the mean density in the simulation cell. **e** Our PIMC simulations give us access to all many-body correlations in the systems, including the spin-resolved electron-electron pair correlation functions $g_{\uparrow\uparrow}(r)$ and $g_{\uparrow\downarrow}(r)$, and the ion-ion pair correlation function $g_{II}(r)$. **f** Electron-ion and ion-ion static structure factors $S_{eI}(q)$ and $S_{II}(q)$, giving us access to the ratio of elastic and inelastic contributions to the full scattering intensity, see the main text; red and blue colors distinguish $T = 100$ eV and $T = 190$ eV.

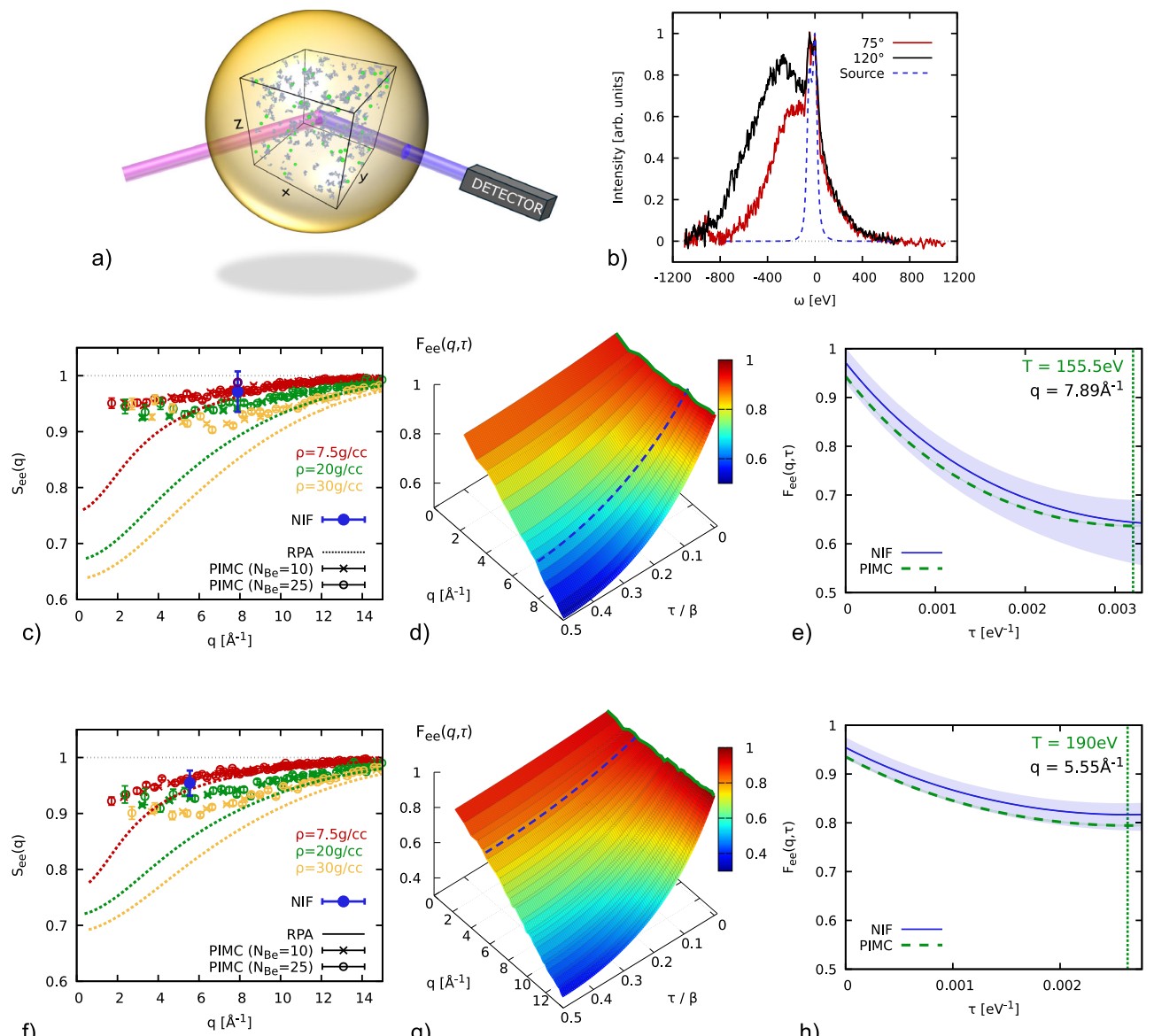

**Fig. 2 | From PIMC to XRTS. a** Schematic illustration of our setup. The Be capsule is compressed and probed by an x-ray source (purple); the scattered photons (blue) are collected by a detector under an angle $\theta$, which is defined as the angle between the purple and blue beams. We use the PIMC method to simulate the quantum degenerate interior of the capsule, allowing for unprecedented comparisons between theory and experiment; **b** XRTS spectra for $\theta = 120°$ (black, dataset #3 in ref. 11) and $\theta = 75°$ (red), and the source-and-instrument function (blue dashed); **c, f** PIMC results for $S_{ee}(q)$ (symbols) for $\rho = 7.5$ g/cc (red), $\rho = 20$ g/cc (green) and $\rho = 30$ g/cc (yellow) compared to the NIF data point (bold blue, with corresponding

uncertainty) and random phase approximation (RPA) results (dotted lines); **d, g** ITCF $F_{ee}(q, \tau)$ in the $q$-$\tau$-plane, with the colored surface and dashed blue line corresponding to PIMC simulations for $\rho = 20$ g/cc and the Laplace transform of the NIF spectra; **e, h** $\tau$-dependence of $F_{ee}(q, \tau)$ at the probed wavenumber; the shaded blue area quantifies the experimental uncertainty. The center and bottom rows correspond to the $\theta = 120°$ and $\theta = 75°$ shots, for which we find $T = 155.5$ eV and $T = 190$ eV (see the vertical dotted lines in (**e, h**), respectively, see the Methods Section.

measurements at 75° and 120°, which we include as the blue crosses. We find very good agreement between PIMC simulation and experimental observation for $\rho = 20$ g/cc for both scattering angles, which is fully consistent with the independent analysis of $S_{ee}(q)$ presented in Fig. 2. To quantify the sensitivity of this analysis on the remaining uncertainty in the inferred temperature, we have carried out additional PIMC simulations at $T = 190 \pm 20$ eV and $T = 155.5 \pm 15$ eV and the corresponding results for $I_{el}/I_{inel}$ are shown as the shaded areas in panels b and c. Evidently, the ratio of the elastic and inelastic contributions is very robust with respect to $\rho$. In particular, the value of $\rho = 34 \pm 4$ g/cc obtained in the original ref. 11 is decisively ruled out.

Let us conclude our analysis of the XRTS data by considering the effective degree of ionization $Z^*$. While its precise value depends on a

particular definition[40], it nevertheless constitutes an often useful concept. Moreover, it is an indispensable ingredient, e.g., for integrated radiation hydrodynamics simulations of fusion experiments and other applications. In Fig. 4, we show the effective degree of ionization as a function of the mass density $\rho$ for $T \sim 155$ eV. The dotted green and dashed blue lines correspond to the Steward-Pyatt model and the popular OPAL EOS, and the black stars to DFT results based on the electrical conductivity[11]. The yellow square shows the original interpretation of the 120° dataset based on the Chihara model by Döppner et al.[11]; it is in good agreement with the DFT curve, but inconsistent with the two other models.

Due to the degree of arbitrariness in its definition, there is no unique way to compute the ionization degree in PIMC; this is in stark

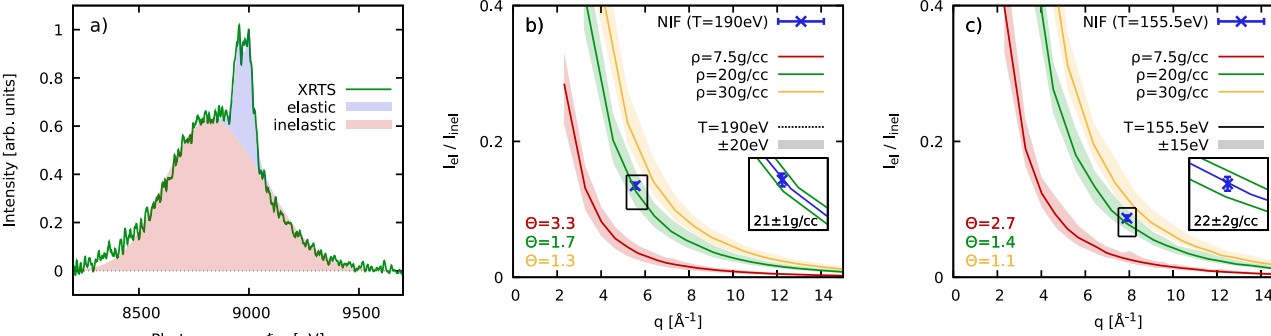

**Fig. 3 | Determination of mass density. a** Data from XRTS measurement on strongly driven Beryllium at $\theta = 75°$ ($q = 5.55\,\text{Å}^{-1}$, green curve) and its elastic (blue) and inelastic (red) contributions; **b, c** wavenumber dependence of the ratio $I_{el}/I_{inel}$ for $\theta = 75°$ [$\theta = 120°$]. Solid lines: PIMC results for $T = 190$ eV [$T = 155.5$ eV] for $\rho = 7.5$ g/cc (red), $\rho = 20$ g/cc (green), and $\rho = 30$ g/cc (yellow); blue cross: NIF measurements [error bars show fit uncertainty following the procedure used by Döppner et al.[11]]; shaded colored areas: PIMC simulations at $T = 190 \pm 20$ eV [$T = 155.5 \pm 15$ eV], addressing the inter-dependence of density and temperature. Corresponding values for the reduced temperature $\Theta = k_B T/E_F$ are given in the bottom left corners. The insets show magnified segments around the NIF data points with the blue and green lines corresponding to PIMC results for $\rho = 21 \pm 1$ g/cc ($\Theta = 1.65 \pm 0.05$) [left] and $\rho = 22 \pm 2$ g/cc ($\Theta = 1.3 \pm 0.1$) [right].

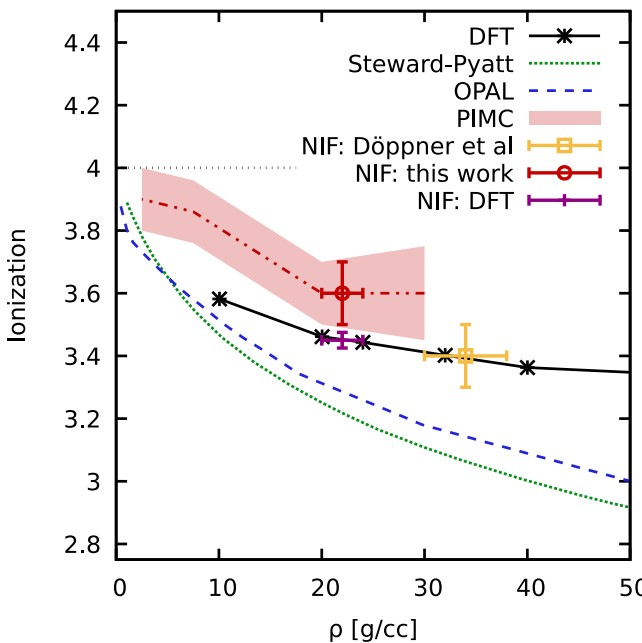

**Fig. 4 | Ionization degree of warm dense Be at $T \sim 155$ eV as a function of the mass density $\rho$.** Red area: present PIMC results [with corresponding uncertainty range], see the Methods section for details; black crosses: DFT-MD taken from ref. 11; dotted green: Steward-Pyatt; dashed blue: OPAL; purple bar: DFT-MD analysis of the Rayleigh weight[31] (see Appendix D of ref. 31 for computational details). Using our new PIMC capabilities to interpret the NIF data at 120° (red circle) substantially changes the inferred parameters compared to the chemical model used in the original ref. 11 (dataset #3).

contrast to all other observables considered in this work. In some situations, cluster analyses based on spatial correlation functions have been successfully employed[41,42], but their application is expected to become increasingly difficult at strong compression when even the orbitals of bound orbitals start to overlap. Moreover, PIMC simulations do not distinguish between bound electrons and the screening cloud. Here, we follow Böhme et al.[43], who suggested to estimate an effective degree of ionization from the electrons capacity to react to an external static perturbation in the linear response regime; see the Methods Section for additional details. The results of this procedure are shown by the red line in Fig. 4 and confirm the underestimation of ionization of the Steward-Pyatt and OPAL models

reported by Döppner et al.[11]. Our final estimate for the conditions of the backscattering XRTS dataset at 120° (75°) is thus given by $T = 155.5 \pm 15$ eV, $\rho = 22 \pm 2$ g/cc, and $Z^* = 3.6 \pm 0.1$ ($T = 190 \pm 20$ eV, $\rho = 21 \pm 2$ g/cc, and $Z^* = 3.88 \pm 0.1$), see the red circle in Fig. 4. Both the temperature and effective ionization agree (within the given uncertainty intervals) with the original study by Döppner et al.[11], whereas the density substantially differs. We further note that a very recent re-analysis of the same XRTS dataset based on DFT-MD results for the Rayleigh weight $W_R(q)$[31], see the Methods Section for additional details, has led to an excellent agreement with the present PIMC results; the correspondingly inferred experimental conditions are depicted by the purple bar in Fig. 4.

This leads us to the following conclusions: ab initio simulations (DFT and PIMC) that do not distinguish between bound and free electrons are in very good agreement, whereas the semi-empirical Chihara model significantly deviates. While it is tempting to attribute the latter to its inherent decomposition, the situation is more complicated as Chihara models entail a host of additional approximations for their individual components. For example, the analysis in the original ref. 11 neglected the stimulated de-excitation of a-priori free electrons into energetically lower bound states (denoted as free-bound transitions[37]) which have been shown to play a significant role at these conditions. Our results thus stress the importance of a holistic and exact treatment of the complicated WDM physics, whereas additional dedicated research will be needed to further understand the source of error in and to potentially improve the computationally cheap Chihara models.

## Discussion

We have presented a framework for the highly accurate ab initio PIMC simulation of warm dense quantum plasmas, treating electrons and ions on the same level. As an application, we have analyzed XRTS measurements of strongly compressed Be using existing data[11] as well as a new data set that probes larger length scales where electronic XC-effects are more important. Due to their unique access to electronic correlation functions, our PIMC simulations have allowed us to independently analyze various aspects of the XRTS signal. This includes the ITCF $F_{ee}(q, \tau)$ and the ratio of elastic to inelastic contributions for which we demonstrated a remarkable consistency between simulation and experiment without the need for any empirical parameters. While the estimation of corresponding electronic pair correlation functions is notoriously difficult for DFT-MD simulations, it is also accessible from restricted PIMC simulations; this might extend the proposed PIMC-based interpretation of XRTS experiments to even lower

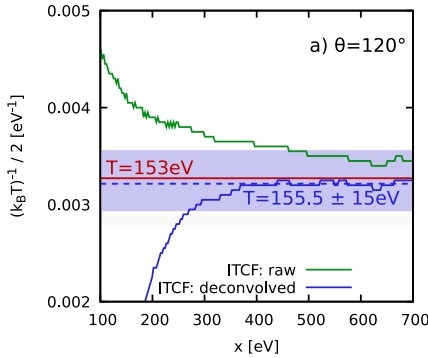

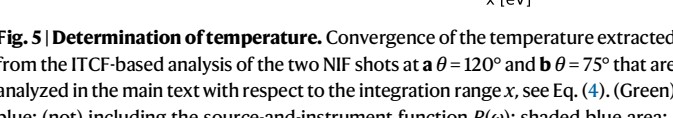

**Fig. 5 | Determination of temperature.** Convergence of the temperature extracted from the ITCF-based analysis of the two NIF shots at **a** $\theta = 120°$ and **b** $\theta = 75°$ that are analyzed in the main text with respect to the integration range $x$, see Eq. (4). (Green) blue: (not) including the source-and-instrument function $R(\omega)$; shaded blue area:

uncertainty of the ITCF analysis. The horizontal red lines in both panels show the temperature estimate from a best fit using improved Chihara models[37] and have been included as a reference.

temperatures and more complicated composite materials that remain inaccessible to exact PIMC simulations due to the sign problem.

Our PIMC simulations accurately capture phenomena that manifest over distinctly different length scales due to the simulation of potentially hundreds of electrons and nuclei[30]. This is particularly important for upcoming XRTS measurements with smaller scattering angles, and for the description of properties that can be probed in the optical limit such as electric conductivity and reflectivity. A key strength is our capability to resolve any type of many-particle correlation function between either electrons or ions. This is in stark contrast with standard DFT-MD simulations, where the computation of electronic pair correlation functions is not possible even if the exact XC functional were known. Moreover, our PIMC simulations are not confined to pair correlation functions and linear response properties such as the dynamic structure factor probed in XRTS[22].

Our highly accurate PIMC results will spark a host of developments in the simulation of WDM. Most importantly, they can unambiguously benchmark the accuracy of existing DFT approaches, and provide crucial input for the construction of advanced nonlocal XC functionals[44]. In addition, our results can quantify potential nodal errors in the restricted PIMC approach, support dynamic methods including time-dependent DFT, and test the basic underlying assumptions in widely used theoretical models.

Finally, our simulations will have a direct impact on nuclear fusion and astrophysics. Due to their dependable predictive capability, they provide both key input for integrated modeling such as transport properties and the equation of state and guide the development of experimental set-ups. A case in point is given by XRTS measurements, for which we have demonstrated the capabilities of our PIMC approach to give a highly consistent interpretation of the data for strongly compressed beryllium. This analysis clearly indicates a substantially lower density compared to previously used chemical models, which has important implications for the interpretation of future experiments, and for integrated radiation hydrodynamics simulations.

Having unraveled electronic correlations in warm dense quantum plasmas, we open the path to study light elements and potentially their mixtures for the extreme conditions encountered during inertial confinement fusion implosions and within astrophysical objects. This will be a true game changer for a field that previously lacked predictive capability.

## Methods
### Model-free analysis of XRTS measurements
To extract the temperature $T$ and normalization $S_{ee}(q) = F_{ee}(q, 0)$ from a measured XRTS signal without the need for models and approximations, we switch to the imaginary-time domain[27,33], where the ITCF $F_{ee}(q, \tau)$ is connected to the dynamic structure factor $S_{ee}(q, \omega)$ via a two-

sided Laplace transform,

$$F_{ee}(q, \tau) = \mathcal{L}\left[S_{ee}(q, \omega)\right] = \int_{-\infty}^{\infty} d\omega\, S_{ee}(q, \omega) e^{-h\omega\tau}. \quad (2)$$

Working in the Laplace domain allows one to separate the physical information of interest from the properties of the combined source-and-instrument function $R(\omega)$ via the well-known convolution theorem[22,34],

$$\mathcal{L}\left[S_{ee}(q, \omega)\right] = \frac{\mathcal{L}\left[S_{ee}(q, \omega) \circledast R(\omega)\right]}{\mathcal{L}[R(\omega)]}. \quad (3)$$

Given accurate knowledge of $R(\omega)$ based on either source monitoring or additional characterization studies[32], it is straightforward to evaluate both the numerator and the denominator of Eq. (3) even in the presence of experimental noise[34].

A particularly useful application of Eqs. (2) and (3) is the symmetry relation of the ITCF, $F_{ee}(q, \beta - \tau) = F_{ee}(q, \tau)$, where $\beta = 1/k_B T$ is the inverse temperature. It is easy to see that the ITCF is symmetric around $\tau = \beta/2$, where it attains a minimum. This symmetry property is equivalent to the detailed balance relation of the dynamic structure factor $S_{ee}(q, -\omega) = e^{-\beta h\omega} S_{ee}(q, \omega)$ that universally holds in thermodynamic equilibrium[45]. This makes it directly possible to infer the temperature from the XRTS signal evaluated in the Laplace domain without any model calculations or approximations.

An additional obstacle is given by the necessarily finite detector range, whereas Eq. (2) in principle requires the integration over an infinite $\omega$-range. In practice, we define the two-sided symmetric incomplete Laplace transform

$$\mathcal{L}_x\left[S_{ee}(q, \omega)\right] = \int_{-x}^{x} d\omega\, S_{ee}(q, \omega) e^{-h\omega\tau}, \quad (4)$$

whose convergence with $x$ is straightforward to check.

In Fig. 5, we show the corresponding ITCF-based temperature analysis of the two XRTS measurements at the NIF. For $\theta = 120°$, we find that the properly deconvolved data (blue) converge around a temperature of $T = 155.5 \pm 15$ eV, in excellent agreement with the improved Chihara model from ref. 37. For $\theta = 75°$, the ITCF analysis gives us a temperature of $T = 190 \pm 20$ eV, with most of the associated uncertainty (shaded blue area) stemming from the influence of the source-and-instrument function in both cases. Following Döppner et al.[11], we only assume a single bulk temperature; this is well justified at the probed time interval as the rebound shock has progressed through the majority of the infalling shell material.

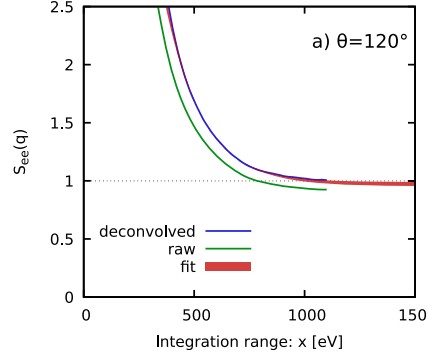
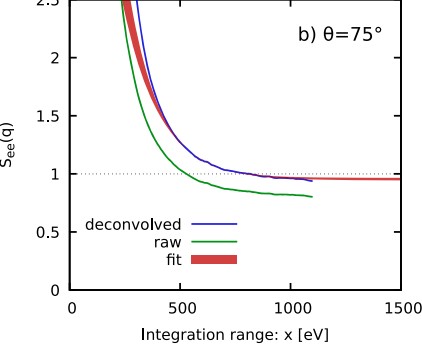

**Fig. 6 | Determination of the normalization.** Convergence of the static structure factor $S_{ee}(q)$ from the imaginary-time f-sum rule following the method introduced in ref. 36 for **a** $\theta = 120°$ and **b** $\theta = 75°$. Blue and green correspond to the properly deconvolved and raw results; the red area shows the extrapolation Eq. (6) and associated uncertainty, and the dotted dark horizontal line corresponds to the uncorrelated value of 1 and serves as a guide to the eye.

Finally, Fig. 6 shows the extraction of the electronic SSF $S_{ee}(q)$, i.e., the proper normalization of the XRTS signal, via the imaginary-time f-sum rule approach, recently introduced in ref. 36; the a-priori unknown normalization constant $C_{norm}$ is inferred from the first derivative of the properly deconvolved ITCF with respect to $\tau$ around the origin via

$$C_{norm} = -\frac{2m_e}{(\hbar q)^2}\frac{\partial}{\partial \tau}\frac{\mathcal{L}[I(q,\omega)]}{\mathcal{L}[R(\omega)]}(\tau = 0).\qquad(5)$$

Figure 6 shows the convergence of the extracted $S_{ee}(q)$ with the symmetrically truncated integration range $x$ for the properly deconvolved (blue) and raw data (green). Evidently, the convergence behavior is well reproduced by the simple exponential ansatz

$$f(x) = A_x + B_x e^{-C_x x},\qquad(6)$$

which is shown as the red curve. It is noted that Eq. (6) is well justified from a theoretical perspective by the expected exponential decay of the dynamic structure factor for large frequencies. In practice, the final estimate for $S_{ee}(q)$ and the corresponding uncertainty (shaded red area in Fig. 6) is obtained by performing several exponential fits over different reasonable intervals $x \in [x_{min}, x_{max}]$ (we use $x_{min} \equiv 500$ eV and $x_{max} = 800$ eV and 1200 eV) and by estimating the spread in the fitting parameter $A_x$.

### Ratio of elastic and inelastic contributions

It is common practice to express the full electronic DSF as a sum of elastic and inelastic contributions[46],

$$S_{ee}(q,\omega) = \underbrace{W_R(q)\delta(\omega)}_{S_{el}(q,\omega)} + S_{inel}(q,\omega).\qquad(7)$$

We note that Eq. (7) does not presuppose any artificial decomposition into effectively bound and free electrons. Instead, the quasi-elastic contribution is due to the longer time scales of the ions due to their heavier mass; it is shaped as the source function $R(\omega)$ in the measured XRTS intensity. From a theoretical perspective, it is straightforward to express the integrated ratio of $S_{el}(q, \omega)$ and $S_{inel}(q, \omega)$ as

$$\frac{I_{el}(q)}{I_{inel}(q)} = \frac{\int_{-\infty}^{\infty} d\omega\, S_{el}(q,\omega)}{\int_{-\infty}^{\infty} d\omega\, S_{inel}(q,\omega)} = \frac{W_R(q)}{S_{ee}(q) - W_R(q)} = \left(\frac{S_{ee}(q)S_{II}(q)}{S_{el}^2(q)} - 1\right)^{-1}.\qquad(8)$$

The theoretical estimation of $I_{el}(q)/I_{inel}(q)$ thus requires explicit simulation results for pair correlation functions (evaluated in Fourier space, to get $S_{ab}(q)$ instead of $g_{ab}(r)$) between all types of particles in the system. This is straightforward for PIMC (and also restricted PIMC), whereas no direct estimation of $S_{ee}(q)$ is possible in DFT-MD. In contrast, the Rayleigh weight $W_R(q) = S_{el}^2(q)/S_{II}(q)$ does not include $S_{ee}(q)$ and is thus more suitable for DFT-MD simulations[46], see also the recent ref. 31 for a corresponding comparison between DFT and PIMC.

### PIMC estimation of ionization degree

To estimate the effective degree of ionization $Z^*$, we follow Böhme et al.[43] and use our new PIMC set-up to solve the electronic problem within the external potential of a fixed ion configuration. Specifically, we apply an additional harmonic external potential of wavenumber $q$ and perturbation amplitude $A$ to the electrons, and study their response in Fourier space. In the limit of infinitesimally weak perturbations, we can expand the density component $\rho_{\mathbf{q}} = \sum_{l=1}^{N} e^{-i\mathbf{r}_l\mathbf{q}}$ as a function of the perturbation amplitude $A$[22,43]

$$\rho_{\mathbf{q}}(A) = \rho_{\mathbf{q}}(0) + A\chi(q) + A^3\chi^{(3)}(q) + \dots,\qquad(9)$$

with $\chi(q)$ being the well-known static linear response function. We further note that the unperturbed component $\rho_{\mathbf{q}}(0)$ vanishes for uniform systems, but is generally non-zero in the inhomogeneous case, i.e., in the presence of fixed nuclei. Moreover, $\chi^{(3)}(q)$ corresponds to the cubic response at the first harmonic[22], which, however, is not of interest for the present work.

In the left panel of Fig. 7, we show $\rho_{\mathbf{q}}$ as a function of the perturbation amplitude $A$ for $N_{Be} = 4$ beryllium atoms for four different wave vectors. We note that every data point has been obtained from an independent PIMC simulation for a particular combination of $\mathbf{q}$ and $A$, making this analysis computationally involved. The dashed black lines show cubic fits according to Eq. (9), which are in excellent agreement with the PIMC results. The dotted red lines show the corresponding linear-response limits (i.e., setting $\chi^{(3)}(q) \equiv 0$). In the right panel, we show the $q$-dependence of the static linear response function, where the green crosses and red circles depict results for $N_{Be} = 4$ and $N_{Be} = 10$ beryllium atoms, respectively; we find no significant finite-size effects, as it is expected at these conditions, see ref. 22 and references therein. The dashed blue line has been obtained for a uniform electron gas (UEG) at the same density and temperature, and would correspond to a fully ionized system, i.e., $Z^* = 4$. However, the true electronic density response is somewhat reduced, as the effectively bound electrons will not respond to a sufficiently weak perturbation.

At this point, we feel that a dedicated discussion of the involved length scales $\lambda = 2\pi/q$ is pertinent. In the limit of $q \to 0$, the density response vanishes due to screening effects, effectively masking any differences between bound electrons and the free electron gas. Conversely, even bound electrons can readily react to an external potential

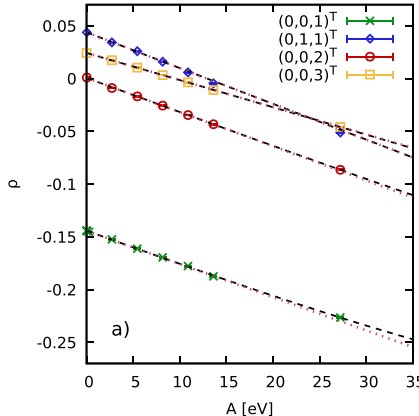
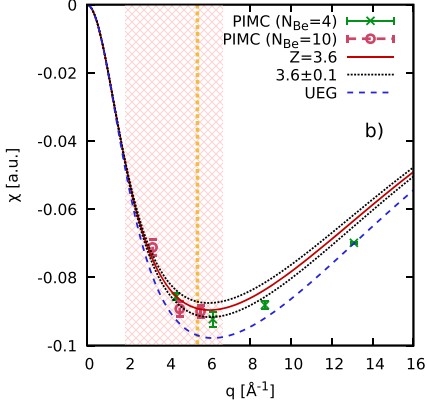

**Fig. 7 | PIMC based estimation of the effective ionization degree $Z^*$ following the approach introduced by Böhme et al.[43]. a** Fourier component of the density $\rho_\mathbf{q}$ [see main text] as a function of the perturbation amplitude $A$ for four different wave vectors $\mathbf{q}$ [error bars show the $1\sigma$ Monte Carlo uncertainty]. The dashed black lines show cubic fits [Eq. (9)], and the dotted red lines the corresponding linear-response limit. **b** Linear density response function $\chi(q)$ for $N_{\mathrm{Be}} = 4$ (green) and $N_{\mathrm{Be}} = 10$ (red) beryllium atoms [error bars show the fit uncertainty in the linear

coefficients]. Dashed blue: density response of a uniform electron gas (UEG) at the same conditions, corresponding to a fully ionized system; solid red: density response of a UEG assuming a fraction of free electrons of 3.6/4. The vertical dotted yellow line indicates the Fermi wavenumber $q_{\mathrm{F}} = (9\pi/4)^{1/3}/r_s$. The red mesh indicates the wavenumber region of interest in which the density response is sensitive to electronic localization around the ions.

in the single-particle limit of $\lambda \ll r_s$; this can be seen for instance in the PIMC data point at $q \approx 14$ Å$^{-1}$ that nearly coincides with the UEG result. To quantify the effective degree of ionization $Z^*$, we thus consider the regime of intermediate wavenumbers $q \lesssim q_{\mathrm{F}}$ (with the Fermi wavenumber $q_{\mathrm{F}}$ being indicated by the vertical yellow line), where $\lambda$ is comparable to the average interparticle distance. Specifically, we match the density response of a UEG with a reduced, effectively free density to the PIMC results; for the present example, we find a best fit at $Z^* = 3.6$, see the solid red curve in Fig. 7.

While this definition of $Z^*$ is, by necessity, somewhat arbitrary, it combines a number of advantages. First, it circumvents the need for a strict distinction between bound and free electrons, which is notoriously difficult in PIMC. Indeed, a cluster analysis that only considers spatial correlations between electrons and nuclei cannot easily distinguish bound electrons from the loosely associated screening cloud, which is particularly problematic at the high densities compared in the present work. Instead, our definition measures the degree of electronic localization around the nuclei. Bound electrons will not respond or respond only very little, respectively, whereas loosely associated but effectively unbound electrons contribute to the full density response despite being located in the vicinity of an ion. Therefore, this procedure automatically takes into account medium effects such as ionization potential depression, without the explicit need to resolve them.

## PIMC simulation details

We consider a fully spin-polarized system where $N_\uparrow = N_\downarrow = N/2$ with $N$ being the total number of electrons. Moreover, we consider effectively charge neutral systems where $N = Z_{\mathrm{tot}} N_{\mathrm{Be}}$ with $Z_{\mathrm{tot}} = 4$ being the atomic charge and $N_{\mathrm{Be}}$ the total number of atoms. The corresponding Hamiltonian governing the behavior of the thus defined two-component plasma then reads

$$\hat{H} = -\frac{\hbar^2}{2m_e} \sum_{l=1}^{N} \nabla_l^2 - \frac{\hbar^2}{2m_I} \sum_{l=1}^{N_{\mathrm{Be}}} \nabla_l^2$$
$$+ e^2 \left\{ \sum_{l<k}^{N} \phi_{\mathrm{E}}(\hat{r}_l, \hat{r}_k) + Z_{\mathrm{tot}}^2 \sum_{l<k}^{N_{\mathrm{Be}}} \phi_{\mathrm{E}}(\hat{I}_l, \hat{I}_k) - Z_{\mathrm{tot}} \sum_{k=1}^{N} \sum_{l=1}^{N_{\mathrm{Be}}} \phi_{\mathrm{E}}(\hat{I}_l, \hat{r}_k) \right\}.$$
(10)

The pair potential $\phi_{\mathrm{E}}(r_a, r_b)$ is given by the usual Ewald sum, where we follow the definitions of Fraser et al.[47].

The basic idea of the PIMC method[15] is to express the canonical partition function (i.e., particle number $N$, volume $V$, and inverse temperature $\beta$ are fixed) in coordinate space,

$$Z_{N,V,\beta} = \frac{1}{N^\uparrow! N^\downarrow!} \sum_{\sigma_{N^\uparrow} \in S_{N^\uparrow}} \sum_{\sigma_{N^\downarrow} \in S_{N^\downarrow}} \xi^{N_{\mathrm{pp}}} \int_V d\mathbf{R} \langle \mathbf{R} \| e^{-\beta\hat{H}} \| \hat{\pi}_{\sigma_{N^\uparrow}} \hat{\pi}_{\sigma_{N^\downarrow}} \mathbf{R} \rangle.$$
(11)

The two sums over all possible permutations $\sigma_{N^\uparrow}, \sigma_{N^\downarrow}$ of the respective permutation groups $S_{N^\uparrow}, S_{N^\downarrow}$ (with $\hat{\pi}_{\sigma_{N^\uparrow}}$ and $\hat{\pi}_{\sigma_{N^\downarrow}}$ realizing a particular permutation) take into account that identical quantum particles cannot be distinguished. We further note that the vector $\mathbf{R}$ contains the coordinates of all electrons and ions in the system. In addition, $N_{\mathrm{pp}}$ is the number of pair exchanges required to realize a particular permutation, and $\xi = 1$, $\xi = 0$, and $\xi = -1$ correspond to the physically meaningful cases of Bose-Einstein, Maxwell-Boltzmann, and Fermi-Dirac statistics, with the latter governing the behavior of the electrons in WDM. For completeness, we note that any exchange effects of the ions can safely be neglected at the conditions that are of interest in the present work.

For $\xi = -1$, it is well known that positive and negative contributions to any observable cancel to a large degree, which leads to an exponential increase in the required compute time with increasing the number of electrons $N = N^\uparrow + N^\downarrow$ or decreasing the temperature $T$; this is the notorious fermion sign problem[16,17]. Here, we follow the approach introduced in refs. 28,29 and consider the general case of a continuous variable $\xi \in [-1, 1]$. The basic idea is to carry out PIMC simulations in the sign-problem free domain of $\xi \geq 0$, and to quadratically extrapolate to the fermionic limit of $\xi = -1$. Indeed, Dornheim et al.[29,30] have shown very recently that this allows for quasi-exact uniform electron gas results for a range of observables including the SSF and ITCF over substantial parts of the WDM regime. A particular strength of this methodology is that it allows one to rigorously assess its accuracy for the case of a small system for which simulations can be performed even in the fermionic limit of $\xi = -1$. Its continued reliability for substantially larger number of particles is then guaranteed both empirically[29], and from the principle of electronic nearsightedness[48]. In essence, this approach allows for unprecedented PIMC simulations of fermions without the exponential computational bottleneck, without any uncontrolled approximations or empirical parameters, and

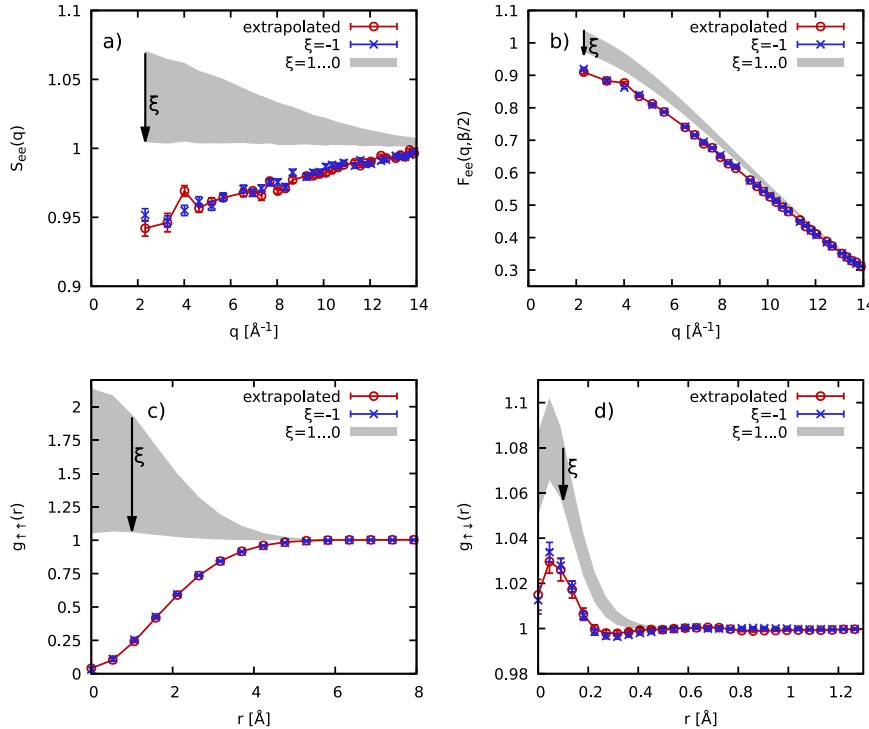

**Fig. 8 | Benchmarking the $\xi$-extrapolation approach for $N_{Be}$ = 10 Be atoms at $T$ = 155.5 eV and $r_s$ = 0.93 ($\rho$ = 7.5 g/cc).** Blue crosses: exact fermionic PIMC results (i.e., $\xi = -1$) [error bars show the $1\sigma$ Monte Carlo uncertainty]; shaded gray area: PIMC results in the sign-problem free domain of $\xi \in [0, 1]$; red circles: quadratic extrapolation to the fermionic limit [error bars show the extrapolation uncertainty]. Panels **a–d** show the electronic static structure factor $S_{ee}(q)$, the thermal static static structure factor $F_{ee}(q, \beta/2)$, the spin-diagonal pair correlation function $g_{\uparrow\uparrow}(r)$ and the spin-offdiagonal pair correlation function $g_{\uparrow\downarrow}(r)$, respectively.

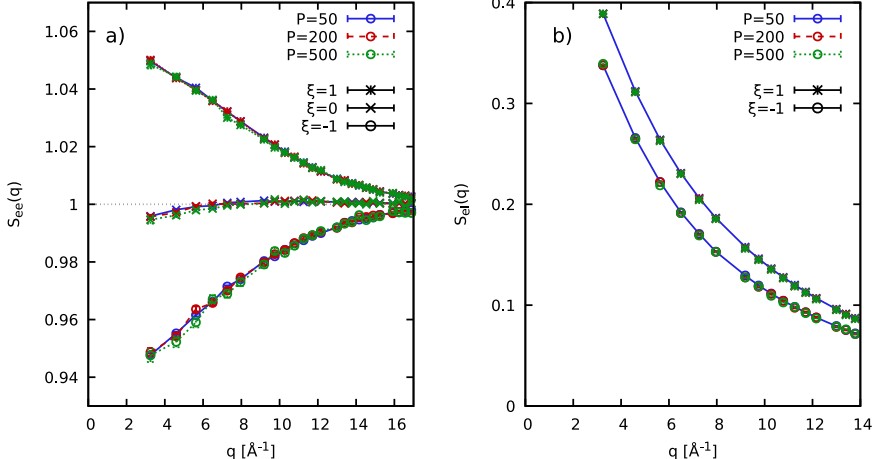

**Fig. 9 | Checking the convergence of the factorization of the density matrix.** Convergence of $S_{ee}(q)$ [**a**] and $S_{eI}(q)$ [**b**] with the number of imaginary-time propagators for $N_{Be} = 4$, $\rho = 7.58$ g/cc, and $T = 155.5$ eV. Error bars show the $1\sigma$ Monte Carlo uncertainty.

without the restrictions on the sampling of the imaginary-time structure inherent to the fixed-node approximation[23].

In Fig. 8, we show this extrapolation for $N_{Be}$ = 10 Be atoms (i.e., $N$ = 40 electrons) at $r_s$ = 0.93 ($\rho$ = 7.5 g/cc) and $T$ = 155.5 eV. The top left panel shows the $q$-dependence of the electronic SSF $S_{ee}(q)$. More specifically, the blue crosses show direct PIMC results for $\xi = -1$ that are subject to the full sign problem. We find an average sign of $S \approx 0.098$, which means that the simulations are computationally involved, but still feasible. In addition, the shaded gray area encompasses our sign-problem free PIMC results for the range $1 \geq \xi \geq 0$. Evidently, the bosonic SSF exhibits the opposite trend compared to the fermions with respect to $q$. Nevertheless, the quadratic extrapolation

that is based on the gray area accurately reproduces the fermionic limit, see the red circles. The same trend is discerned for the thermal structure factor $S_{ee}^{\beta/2}(q) = F_{ee}(q, \beta/2)$, as shown in the top right panel of Fig. 8, although the effects of quantum statistics are less pronounced for $\tau = \beta/2$.

In the bottom row, we repeat this analysis for the spin-resolved electronic PCF, with the left and right panels showing results for the spin-diagonal and spin-offdiagonal component. For $g_{\uparrow\uparrow}(r)$, spin effects predominantly shape the behavior for $r \lesssim 4$ Å; fermions exhibit the familiar exchange–correlation hole with $g_{\uparrow\uparrow}(0) = 0$, whereas bosons tend to cluster around each other exhibiting the opposite trend. Nevertheless, the extrapolation works exceptionally well and

reproduces the fermionic curve. For $g_{\uparrow\downarrow}(r)$, we find a more subtle behavior and spin-effects play a less important role. At the same time, the $\xi$-extrapolation method cannot be distinguished from the fermionic benchmark results within the (small) Monte Carlo error bars.

### Convergence with number of imaginary-time propagators

An indispensable ingredient to the PIMC method is given by the factorization of the density operator $e^{-\beta \hat{H}} \neq e^{-\beta \hat{W}} e^{-\beta \hat{K}}$, where $\hat{W}$ and $\hat{K}$ are the potential and kinetic energy operators. Here, we use an implementation of the pair approximation, as described in ref. 49, that becomes exact in the limit of $P \to \infty$ as $\sim 1/P^4$, where $P$ is the number of imaginary-time propagators, or, equivalently, the number of high-temperature factors.

In Fig. 9, we show results for the electronic SSF $S_{ee}(q)$ [left] and the electron-ion SSF $S_{el}(q)$ [right] for $N_{Be} = 4$ Be atoms at $r_s = 0.9$ ($\rho = 7.58$ g/cc) and $T = 155.5$ eV. In particular, the stars, crosses, and circles correspond to the relevant cases of $\xi = 1$, $\xi = 0$, and $\xi = -1$, respectively, and the blue, red, and green color distinguishes simulation results for $P = 50$, $P = 200$, and $P = 500$. Evidently, we find no systematic dependence even for $P = 50$. In practice, we use $P = 200$ throughout this work, as this has the beneficial side effect of a good $\tau$-resolution for the ITCF $F_{ee}(q, \tau)$.

## Data availability

The PIMC data generated in this study and the XRTS data for $\theta = 75°$ analyzed here have been deposited in the Rossendorf Data Repository (RoDaRe) database under accession code https://doi.org/10.14278/rodare.3673.

## Code availability

All PIMC results that have been presented in this work have been computed using the open-source ISHTAR code that is freely available at ref. 50.

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

## Acknowledgements

We gratefully acknowledge helpful feedback by Mandy Bethkenhagen. This work was partially supported by the Center for Advanced Systems Understanding (CASUS) which is financed by Germany's Federal Ministry of Education and Research (BMBF) and by the Saxon state government out of the State budget approved by the Saxon State Parliament. This work has received funding from the European Research Council (ERC) under the European Union's Horizon 2022 research and innovation programme (Grant agreement No. 101076233, "PREXTREME"). Views and opinions expressed are however those of the authors only and do not necessarily reflect those of the European Union or the European Research Council Executive Agency. Neither the European Union nor the granting authority can be held responsible for them. The work of Ti.D., M.P.B, M.J.M., and F.R.G. was performed under the auspices of the U.S. Department of Energy by Lawrence Livermore National Laboratory under Contract No. DE-AC52-07NA27344. Ti.D., M.P.B., M.J.M., and D.O.G. were supported by Laboratory Directed Research and Development (LDRD) Grant Nos. 24-ERD-044 and 25-ERD-047. The work of L.B.F. is supported by the DOE Office of Science, Fusion Energy Science under FWP 100866, and supported by the Department of Energy, Laboratory Directed Research and Development program at SLAC National Accelerator Laboratory, under contract DE-AC02-76SF00515. The work of T.G. was supported by the European Union's Just Transition Fund (JTF) within the project *Röntgenlaser-Optimierung der Laserfusion* (ROLF), contract number 5086999001, co-financed by the Saxon state government out of the State budget approved by the Saxon State Parliament. The PIMC calculations were carried out at the Norddeutscher Verbund für Hoch- und Höchstleistungsrechnen (HLRN) under grant mvp00024, on a Bull Cluster at the Center for Information Services and High Performance Computing (ZIH) at Technische Universität Dresden, and on the HoreKa supercomputer funded by the Ministry of Science, Research and the Arts Baden-Württemberg and by the Federal Ministry of Education and Research.

## Author contributions

To.D. developed the main idea, carried out all simulations, and wrote substantial parts of the ms. Ti.D., L.B.F., and M.J.M. were involved in the experimental measurement, contributed to the analysis, and to writing the ms. P.T., D.O.G., and J.V. contributed to the analysis, and contributed substantial parts to the ms. M.P.B., T.G., F.R.G., D.K., Zh.A.M., and S.S. contributed to the analysis, and to writing the ms.

## Funding

## Competing interests

The authors declare no competing interests.
