## [Transparent Peer Review file · Nature Communications]

Unraveling electronic correlations in warm dense quantum plasmas

Corresponding Author: Dr Tobias Dornheim

Version 0:

Reviewer comments:

Reviewer #1

(Remarks to the Author)

Accurate diagnostics of warm dense matter (WDM) are crucial for high energy density physics but present significant challenges for both theory and experiments. In this work, Dornheim, Döppner, and co-workers apply path-integral Monte Carlo (PIMC) to analyze recent and new x-ray Thomson scattering (XRTS) data of beryllium under WDM conditions. Their results show consistency between PIMC prediction and XRTS measurements, specifically for the imaginary-time correlation function $F_{ee}(q,\tau)$ and the ratio of elastic to inelastic scattering I_{el}/I_{inel} . These findings have the potential to be both interesting and impactful, provided they are sufficiently well justified.

However, this work is incomplete, as it lacks significant details necessary to clarify key relationships: (1) the connection between the results of this re-analysis and previous analysis of the XRTS data regarding the ρ - T condition, (2) the comparison between the current and previous restricted-node PIMC predictions about I_{el}/I_{inel} , and (3) the relationship between the present PIMC and previous DFT-MD predictions about Z^* (the average charge state).

Specifically,

(1) The re-analysis yielded a ρ - T condition of 20 g/cc and 155 eV, which significantly differs from all four data sets listed in Table 1 of Ref. [11] (Döppner et al., Nature 2023). It is unclear which of the four data sets was re-analyzed, what caused this discrepancy, and what the implications are for the remaining three data sets. The experimental analysis in [11] was conducted using a "widely used approach that has been successfully applied for analysing previous XRTS measurements". If the new analysis is correct, this raises an important question: what was wrong with the previous approach? These are critical issues that the authors must address.

(2) the use of PIMC simulation for XRTS analysis based on $F_{ee}(q,\tau)$ is not novel, as it was previously published in [31] (Dornheim, et al., Nature Communications. 2022). Additionally, correlation functions $g(r)$ between all types of particles have been calculated and published for various materials by the Militzer group using restricted-node PIMC (see, e.g., Militzer & Driver, PRL 2015; Driver et al., J. Chem. Phys. 2015 and High Energy Density Physics 2017; Zhang et al., High Energy Density Physics 2016). These results also allow the calculation of I_{el}/I_{inel} (using Eq. 7). The restricted-node PIMC method has been highly successful in predicting the equations of state for moderately high- Z materials, some of which (e.g., CH along the Hugoniot) have already been experimentally verified (compare Zhang et al., JCP2017 with Döppner et al., PRL2018, and Kritcher et al., Nature 2020). However, the $g(r)$ results for spin-up and down electrons presented in this paper (Fig. 1e) differ significantly from the trends expected based on the Militzer group's results. The authors have neither mentioned this discrepancy nor provided an explanation. It would be theoretically interesting if the authors could offer more detailed comparisons, potentially adding restricted-node PIMC results for $g(r)$ and I_{el}/I_{inel} . This would help demonstrate whether nodal restriction is inferior in predicting key XRTS-derived quantities, if that is indeed the case.

(3) an important conclusion in Ref.[11] was the agreement between experimentally inferred Z^* and DFT-MD simulations. However, given the significant difference in the ρ - T condition derived in this work compared to Ref. [11], the value of Z^* must also be re-evaluated. The authors might consider following the approach by Militzer et al., which estimates Z^* by comparing the integrated nucleus-electron pair correlation function (see references mentioned earlier). Currently, the manuscript includes only a brief statement ("substantial presence of ions with a fully occupied K-shell at 100 eV" on page 5) addressing this important issue. However, this claim is not adequately supported and appears to contradict all other simulation and theoretical models shown in Fig. 3a of [11]. The authors should provide further justification for this claim or clarify the discrepancy.

In addition to the major concerns mentioned above, I have identified the following areas that need further clarification and improvement:

- (1) A colorbar is needed in Fig. 1 c)-d) to quantify the electron density results.
- (2) Please explain how the angle θ is defined in the experiment, perhaps by adding a note to Fig. 2 a). Additionally, the shaded areas in Fig. 2(e)-(f) need to be explained.
- (3) There seem to be some logical issues and inconsistencies regarding Figs. 2 and 3. In Fig. 2 c) and f), the experimental data are shown to be consistent with PIMC for $\rho \leq 20$ g/cc, but all subsequent comparisons are based on PIMC results at $\rho = 20$ g/cc. However, the results shall depend on both ρ and T . How have the authors ruled out other possibilities, such as lower densities and temperatures, or higher densities and temperatures? Without this information, the "excellent agreement" observed at 20 g/cc seems coincidental, raising concerns about whether the approach can be extended to other experimental shots or materials to reliably determine T and ρ .
- (4) Can the authors include uncertainties for their various PIMC results?
- (5) Can the authors provide more experimental details regarding the new data to justify that the density is indeed 20 g/cc and is the same as the previous shot (which specific shot from [1 1] is being referred to)?
- (6) The following two statements needs revision: "... standard DFT-MD simulations, where the computation of electronic correlation functions is not possible even if the exact XC functional were known" (What is the intended point here? An "exact XC function" does not exist within the single-particle approximation of DFT. However, if an exact XC were known, then the correlation would also be known simply by subtracting the exact exchange term); "our PIMC simulations are not confined to two-particle correlation functions and linear response properties ..." (Do you mean some other properties can also be calculated? If so, please specify what those properties are).
- (7) In Fig. 5: there is no "dotted red curve" as mentioned. What do the dotted black lines represent?
- (8) On page 12: why is the Rayleigh weight "thus more suitable" for DFT-MD simulations? Please clarify the reasoning.
- (9) In Eq. 9: please explain what σ_N and S_N represent.
- (10) instead of merely claiming "our results can quantify the nodal error in the restricted PIMC approach", this should be explicitly addressed in the research. I encourage the authors to quantify and clearly demonstrate whether the node error has a significant impact, and if so, how much it matters.

Reviewer #2

(Remarks to the Author)

The manuscript titled "Unraveling Electronic Correlations in Warm Dense Quantum Plasmas" presents significant progress in the field of plasma physics, specifically regarding the challenging theoretical description of WDM. The authors introduce a novel path integral Monte Carlo approach that bypasses nodal restrictions, enabling a more comprehensive exploration of electronic correlations in light elements under extreme conditions. Their application to strongly compressed beryllium and the comparison with experimental X-ray Thomson scattering data from the National Ignition Facility are particularly compelling. The study's findings demonstrate excellent agreement between simulation and experiment, underscoring the reliability of the method without reliance on empirical parameters. Overall, this work contributes valuable insights into electronic structures in WDM, but it is somewhat questionable to publish it in Nature Communication unless they provide strong answers to my questions below and corresponding revision on the manuscript:

A) The key point of this study is that it resolves the issue of the traditional PIMC method being underutilized in warm dense matter (WDM) studies due to challenges such as the fermion sign problem, by employing the novel approach of "carrying out a controlled extrapolation that is substituted into the canonical partition function." However, this methodology has already been presented in detail in the author's previously published references [28] and [29], and is thoroughly described in reference [35], which diminishes the originality of the paper.

B) Another aspect of novelty lies in the application of this methodology to X-ray Thomson scattering data from the National Ignition Facility (NIF). However, this too has been extensively discussed in reference [32], further reducing the originality of the work.

Therefore, to emphasize the novelty of the paper, I believe it would be much more effective if the overall narrative were refocused on the new physical insights embedded in the NIF beryllium data that had not been discovered previously.

C) Additionally, although this may be challenging due to the limitations of the NIF experiment, the experimental data seems insufficient to fully support the theoretical claims. In particular, in Figures 2(c) and (f) and Figure 3, there is only one experimental data point for each angle. It would be helpful to provide additional experimental data points or related experimental results to more convincingly validate the theory.

D) Other minor comments and questions are as follows:

1. What does "XC" stand for? If it refers to Exchange Correlation, it would be helpful to clarify this for non-experts.
2. It is impressive that a model-free temperature estimation can be made using ITCF. What is the uncertainty associated with this method for estimating the temperature (for each angle)? How many significant figures can be reliably reported? Furthermore, how does this relate to the temperature distribution observed in the experiment?
3. What is the density of the NIF data in Figure 2?
4. Figure 4: Why was the Chihara model chosen for comparison? Is this model regarded as the best-known standard?
 - o Although the data falls within the uncertainty range, why do the Chihara model and the author's model show greater discrepancies at small angles? Is it because small angles are sensitive to electronic correlations? If so, which specific physical factors determining the correlation function are particularly sensitive, leading to such discrepancies? How does the Chihara model treat those physical factors?
5. What does "W" in Equation 8 refer to? If it is the Ewald potential, please state this explicitly. Additionally, if possible, I

suggest altering the notation, as it could be confused with the Rayleigh weight.

Reviewer #3

(Remarks to the Author)

This paper applies a recently developed method for simulation of relatively large numbers of Fermionic quantum particles. Typically this is accomplished using Kohn-Sham density functional theory (DFT), but this introduces a variety of well-known limitations including an uncontrolled approximation for the exchange-correlation functional and the Born-Oppenheimer approximations. The alternative path-integral Monte Carlo (PIMC) method avoids these limitations, but at a computational cost that scales exponentially in the number of fermions. Building on the original idea of bosonic to fermionic extrapolation (Xiong and Xiong, 2022), the authors have developed a practical implementation of the method for simulating strongly coupled electron-ion fluids at high temperature a.k.a. warm dense matter (WDM). Using careful benchmarking against exact PIMC results for small systems, they show that the extrapolation accuracy is very good and insensitive to system size. They subsequently apply the method to calculate the spatial and temporal correlations of electrons and ions in beryllium at several WDM conditions. They are able to use these results to directly compare with x-ray Thomson scattering (XRTS) data collected from shots at the National Ignition Facility

The work is highly original and the quality is very high. I anticipate that this paper will become highly cited for two reasons:

1. It establishes the validity and usefulness of an important new method for directly simulating warm dense matter
2. It provides insight in to recent experiments at NIF that are of fundamental scientific importance in astrophysics and other fields.

Version 1:

Reviewer comments:

Reviewer #1

(Remarks to the Author)

The authors have significantly improved the manuscript's quality and clarity, addressing all my previous comments satisfactorily.

However, I still notice the following issues that need to be corrected in the revised manuscript:

1. On page 3, the example and reference in "...and experimentally verified, e.g., along the Hugoniot of Si [26]" are not accurate. A better example would be CH (Zhang et al., JCP2017), which was verified by the experiments of Döppner et al., PRL2018, and Kritcher et al., Nature 2020.
2. In the legend of Fig. 3, the theta values, instead of T, should be used to accurately label the NIF data (blue crosses) to reflect that the temperature in those experiments was not constrained to those values until this research.
3. In Fig. 7 and the corresponding discussions, it would be better to use "eV" and " A^{-1} " for the units of A and q, respectively, to maintain consistency with other parts of the manuscript.
4. In the caption of Fig. 7, "a free fraction of electrons" should be changed to "a fraction of free electrons."
5. The legend for PIMC data in Fig. 7 should be updated to "PIMC ($N_{Be}=4$)" for green crosses and "PIMC ($N_{Be}=10$)" for red circles for clarity.
6. On page 15, "as these conditions" should be changed to "at these conditions"; also, there is a redundant word in "not not respond to."

Reviewer #2

(Remarks to the Author)

This study employs a novel ab initio Path Integral Monte Carlo (PIMC) simulation to analyze two X-ray Thomson Scattering (XRTS) datasets of strongly compressed beryllium obtained from the National Ignition Facility (NIF). The authors have adequately addressed my previous concerns regarding the novelty of their methodology, convincingly demonstrating that this is the first application of PIMC (including restricted PIMC, ξ -extrapolation, or other variants) to the interpretation of XRTS experiments. This represents a substantial advancement, as it highlights the potential of establishing robust ab initio simulations for interpreting such data. Furthermore, the additional PIMC calculations carried out by the authors have provided a novel approach to reassess the validity of commonly used EOS models, such as Steward-Pyatt and OPAL, as employed in the original work by Döppner et al. This contribution is particularly valuable as it offers an opportunity to re-evaluate existing models and experimental interpretations, paving the way for significant advancements in future studies. In conclusion, this manuscript demonstrates both academic originality and practical contributions, making it a strong candidate

for publication in your journal. I wholeheartedly recommend its acceptance.

Response to referee reports: “Unraveling electronic correlations in warm dense quantum plasmas”

T. Dornheim *et al.*

January 27, 2025

Dear Editors,

we would like to thank all referees for their thorough reading of our work, and the positive and constructive feedback. In response, we have carried out extensive additional PIMC simulations and substantially revised parts of the ms. Please find a detailed list of all changes below. For the referees' convenience, we also attach a diff file highlighting all modifications.

We hope that with these changes, our work is now suitable for publication in *Nature Communications*.

Sincerely,

T. Dornheim, for the authors

Comments by Referee 1

Accurate diagnostics of warm dense matter (WDM) are crucial for high energy density physics but present significant challenges for both theory and experiments. In this work, Dornheim, Döppner, and co-workers apply path-integral Monte Carlo (PIMC) to analyze recent and new x-ray Thomson scattering (XRTS) data of beryllium under WDM conditions. Their results show consistency between PIMC prediction and XRTS measurements, specifically for the imaginary-time correlation function $F_{ee}(q, \tau)$ and the ratio of elastic to inelastic scattering I_{el}/I_{inel} . These findings have the potential to be both interesting and impactful, provided they are sufficiently well justified.

We thank the referee for reviewing our work, and for this positive assessment.

However, this work is incomplete, as it lacks significant details necessary to clarify key relationships: (1) the connection between the results of this re-analysis and previous analysis of the XRTS data regarding the ρ - T condition, (2) the comparison between the current and previous restricted-node PIMC predictions about I_{el}/I_{inel} , and (3) the relationship between the present PIMC and previous DFT-MD predictions about Z^* (the average charge state).

Specifically, (1) The re-analysis yielded a ρ - T condition of 20 g/cc and 155 eV, which significantly differs from all four data sets listed in Table 1 of Ref. [11] (Döppner et al., Nature 2023). It is unclear which of the four data sets was re-analyzed, what caused this discrepancy, and what the implications are for the remaining three data sets. The experimental analysis in [11] was conducted using a "widely used approach that has been successfully applied for analysing previous XRTS measurements". If the new analysis is correct, this raises an important question: what was wrong with the previous approach? These are critical issues that the authors must address.

The backscattering XRTS dataset at 120° that has been re-analyzed in the present work corresponds to dataset #3 in the original Nature paper by Döppner et al; this is now made explicitly clear both in figure captions and the corresponding discussions in the main text.

The source of error in the Chihara model used in the original Nature paper remains yet unresolved. A subsequent DFT-based re-analysis of the same XRTS dataset (analyzing the Rayleigh weight $\tilde{W}_R(q)$, Ref. [31] in the new ms) has revealed excellent agreement to our PIMC results. Since neither PIMC nor DFT-MD inherently distinguish between bound and free electrons, it would be tempting to attribute the observed discrepancy of the Chihara model to this decomposition. Yet, we feel that such a conclusion would be pre-mature, as there is a host of potential additional sources of error in the Chihara models; for example, the model by Döppner *et al.* violates the *detailed balance* relation between energy gain and energy loss of the scattered photon by not considering the de-excitation of a-priori free electrons into energetically lower bound states. **We agree that these considerations have important implications for future works and have added a corresponding discussion to the main text of revised ms.**

(2) the use of PIMC simulation for XRTS analysis based on $F_{ee}(q, \tau)$ is not novel, as it was previously published in [31] (Dornheim, et al., Nature Communications. 2022). Additionally, correlation functions $g(r)$ between all types of particles have been calculated and published for various materials by the Militzer group using restricted-node PIMC (see, e.g., Militzer & Driver, PRL 2015; Driver et al., J. Chem. Phys. 2015 and High Energy Density Physics 2017; Zhang et al., High Energy Density Physics 2016). These results also allow the calculation of I_{el}/I_{inel} (using Eq. 7). The restricted-node PIMC method has been highly successful in predicting the equations of state for moderately high- Z materials, some of which (e.g., CH along the Hugoniot) have already been experimentally verified (compare Zhang et al., JCP2017 with Döppner et al., PRL2018, and Kritcher et al., Nature 2020). However, the $g(r)$ results for spin-up and down electrons presented in this paper (Fig. 1e) differ significantly from the trends expected based on the Militzer group's results. The authors have neither mentioned this discrepancy nor provided an explanation. It would be theoretically interesting if the authors could offer more detailed comparisons, potentially adding restricted-node PIMC results for $g(r)$ and I_{el}/I_{inel} . This would help demonstrate whether nodal restriction is inferior in predicting key XRTS-derived quantities, if that is indeed the case.

Figure 1: Spin-resolved pair correlation function of hydrogen at $r_s = 10$ and $T = 31,250$ K. The green lines show restricted PIMC results by Militzer and Ceperley [Phys. Rev. E **63**, 066404 (2001)], and the red symbols the ξ -extrapolation method utilized in the present work.

While the ITCF $F(q, \tau)$ has indeed been used in previous works to interpret XRTS signals, all of these efforts were **simulation-free**; they relied e.g. on the symmetry relation $F(q, \tau) = F(q, \beta - \tau)$ to infer the inverse temperature $\beta = 1/k_B T$ or on the f-sum rule that determines the first derivative of $F(q, \tau)$ with respect to τ , but did not require any model or simulation data for $F(q, \tau)$. **The present work thus constitutes the first direct comparison between PIMC results for $F(q, \tau)$ and the experimental observation.** We have added a corresponding explanation to the discussion of Fig. 2 in the main text.

We thank the referee for pointing out previous works centered on the restricted PIMC (RPIMC) method. It has not been our aim to devalue previous RPIMC simulations, or to claim an inferior performance of RPIMC at the present conditions. We have substantially revised various parts of the ms to more clearly put RPIMC and our present work into the correct perspective by emphasizing strengths and weaknesses of both methods, clearly delineating differences, and pointing out future perspectives for both. This includes:

- We added references to previous successful comparisons between RPIMC and Hugoniot measurements.
- We clearly state that RPIMC does entail a fixed nodal structure, which our method does not; at the same time, we do not wish to speculate about the magnitude of potential nodal errors at the present conditions. Note that we do not have any experience with RPIMC ourselves and do not have access to a corresponding implementation. We do hope to compare our simulations with corresponding calculations at the same conditions by other experts in the field, similar to what we have done for the more simple uniform electron gas model system together with Militzer [see, e.g., PRB **103**, 205142 (2021)].
- A key difference between RPIMC and our approach is that the former—independent of any hypothetical nodal errors—does not give one access to the ITCF $F(q, \tau)$. This gives us more options to compare with XRTS experiments compared to RPIMC.
- **We are not aware that RPIMC (or any other PIMC method, for that matter) had ever been employed before to interpret an XRTS experiment. Therefore, our work opens up exciting new avenues to use RPIMC to interpret XRTS measurements by considering frequency-integrated correlation functions, such as $S_{ee}(q)$, I_{el}/I_{inel} , or the Rayleigh weight $W_R(q)$; this promising potential is made explicitly clear in the discussion section of the revised ms.**

Finally, we have carried out additional PIMC calculations for hydrogen at $T = 31, 250$ K and $r_s = 10$ to compare our spin-resolved pair correlation functions with corresponding RPIMC data by Militzer and Ceperley, see Fig. 1. These conditions have been selected for this comparison as they exhibit a similar trend for $g_{\uparrow\downarrow}(r)$ as observed for Beryllium at $T = 100$ eV and $\rho = 7.5$ g/cc shown in Fig. 1 e) in the main text where a spin-up and a spin-down electrons tend to cluster around an ion in some configurations. Evidently, we find excellent agreement between the present methodology and existing RPIMC results from the literature; the small discrepancies are likely an artifact due to the finite resolution of the plot in [Militzer and Ceperley, Phys. Rev. E **63**, 066404 (2001)] from where we have extracted the data.

(3) an important conclusion in Ref. [11] was the agreement between experimentally inferred Z^* and DFT-MD simulations. However, given the significant difference in the ρ - T condition derived in this work compared to Ref. [11], the value of Z^* must also be re-evaluated. The authors might consider following the approach by Militzer et al., which estimates Z^* by comparing the integrated nucleus-electron pair correlation function (see references mentioned earlier). Currently, the manuscript includes only a brief statement ("substantial presence of ions with a fully occupied K-shell at 100 eV" on page 5) addressing this important issue. However, this claim is not adequately supported and appears to contradict all other simulation and theoretical models shown in Fig. 3a of [11]. The authors should provide further justification for this claim or clarify the discrepancy.

We agree with the referee that the important issue of ionization was not adequately addressed in the ms. Indeed, while often useful, the concept of "effective ionization" is somewhat arbitrary and depends on a particular definition and/or a particular employed model. We appreciate the referee's suggestion to employ a cluster analysis. At the same time, we feel that some of the underlying assumptions are best suited for low densities, but might become problematic at strong compression when even the orbitals of effectively bound electronic states start to overlap.

To adequately address the question of ionization with PIMC, we have carried out very extensive additional simulations following the idea by Böhme *et al.* [PRL **129**, 066402 (2022)], who suggested to estimate an effective fraction of free electrons by applying an external harmonic perturbations of a suitable wavenumber to the electrons while keeping the nuclei fixed. While being computationally expensive due to the need to carry out independent PIMC simulations for different wavenumbers, perturbation amplitudes and, in the present case, different values of the spin-statistics variable ξ , this method has a number of advantages. In particular, it does not presuppose a binary distinction between bound electrons and a more loosely associated screening cloud. Instead, it measures the (continuous) capacity of any electron to react to the external potential, which is maximal for free electrons, and can be diminished but non-zero in the vicinity of a nucleus; it will only vanish when the electron is indeed tightly bound.

The results are discussed in the new Fig. 4, where we compare our new PIMC results to a number of previous models (Steward-Pyatt and OPAL) and simulations (DFT-MD) considered by Döppner *et al.*. Overall, our new PIMC results confirm the trend observed in the original Nature paper, namely an underestimation of Z^* by the two aforementioned models in particular at strong compression. Indeed, our estimate of $Z_{\text{PIMC}}^* = 3.6 \pm 0.1$ for the 120° dataset agrees with the result by Döppner *et al.*, $Z_{\text{Chihara}}^* = 3.4 \pm 0.1$, within error bars. Moreover, we substantiate the DFT-MD results also published in the Döppner *et al.* paper; the slightly higher Z^* from PIMC might simply be a matter of definition since we, as discussed above, do not impose a binary distinction between *bound* and *free* but have a non-zero contribution from the screening cloud.

The main difference with respect to the original Nature paper in the interpretation of XRTS dataset #3 is the lower mass density of $\rho = 22 \pm 2$ g/cc (compared to the Chihara-based estimate of $\rho = 34 \pm 4$ g/cc); this result has subsequently been further substantiated by independent DFT-MD simulations (see Ref. [31] in the new ms) based on an analysis of the Rayleigh weight $W_R(q)$. The

corresponding DFT-MD estimate for the ρ - Z^* -state point has also been included in the new Fig. 4 and is consistent with the present PIMC-based analysis.

These points are discussed extensively in the new version of the ms, which also includes a new subsection in the Methods Section about the PIMC based estimation of Z^* . Furthermore, we have re-phrased the sentence "substantial presence of ions with a fully occupied K-shell at 100 eV" on page 5 to "indicates the onset of clustering of two electrons around a single nucleus at the lower temperature".

In addition to the major concerns mentioned above, I have identified the following areas that need further clarification and improvement: (1) A colorbar is needed in Fig. 1 c)-d) to quantify the electron density results.

We have included a colorbar as requested.

(2) Please explain how the angle θ is defined in the experiment, perhaps by adding a note to Fig. 2 a). Additionally, the shaded areas in Fig. 2(e)-(f) need to be explained.

We have added a clearer definition of the angle θ in the caption of Fig. 2 as suggested. Moreover, we now explain that the shaded areas indicate the uncertainty in the experimental results.

(3) There seem to be some logical issues and inconsistencies regarding Figs. 2 and 3. In Fig. 2 c) and f), the experimental data are shown to be consistent with PIMC for $\rho \leq 20$ g/cc, but all subsequent comparisons are based on PIMC results at $\rho = 20$ g/cc. However, the results shall depend on both ρ and T . How have the authors ruled out other possibilities, such as lower densities and temperatures, or higher densities and temperatures? Without this information, the "excellent agreement" observed at 20 g/cc seems coincidental, raising concerns about whether the approach can be extended to other experimental shots or materials to reliably determine T and ρ .

To address this very important point concerning the inter-dependence of density and temperature, we have carried out additional PIMC simulations at $T = 155.5 \pm 15$ eV and $T = 190 \pm 20$ eV. The corresponding results for the ratio of elastic to inelastic scattering I_{el}/I_{inel} have been included as the shaded area in the now extended Fig. 3. In practice, the cross-dependence is small, and the mass density of $\rho = 34 \pm 4$ g/cc that has been found in the original paper by Döppner *et al.* for the 120° backscattering dataset remains inconsistent with our PIMC results even within the possible margins of the temperature.

(4) Can the authors include uncertainties for their various PIMC results?

All PIMC results shown in this work have been included with the corresponding Monte Carlo error bars, which are, however, often small and, therefore, sometimes not clearly visible. To further transparency and increase the impact and reproducibility of our results, we will make all data available in a publicly accessible repository upon publication.

(5) Can the authors provide more experimental details regarding the new data to justify that the density is indeed 20g/cc and is the same as the previous shot (which specific shot from [11] is being referred to)?

It can be assumed that the plasma conditions probed in the two measurements are comparable as (1) the probe times are identical within the timing uncertainty ($\Delta t = 0.05$ ns) and (2) the shot-to-shot fluctuations are small and carefully controlled by high-accuracy laser delivery and high reproducibility in target production.

We have added this information to the discussion of Fig. 2 in the main text.

(6) The following two statements needs revision: "... standard DFT-MD simulations, where the computation of electronic correlation functions is not possible even if the exact XC functional were known" (What is the intended point here? An "exact XC functional" does not exist within the single-particle approximation of DFT. However, if an exact XC were known, then the correlation would also be known simply by subtracting the exact exchange term); "our PIMC simulations are not confined to two-particle correlation functions and linear response properties ..." (Do you mean some other properties can also be calculated? If so, please specify what those properties are).

We apologize for this ambiguous wording. DFT requires the so-called exchange–correlation (XC) functional as an external input. In principle, there exists an exact XC-functional that would make DFT—despite its inherent single-electron picture—exact (at least within the usual Born-Oppenheimer approximation, which, however, is not expected to introduce any additional errors in the considered parameter regime). That being said, even such exact DFT simulations would only give one access to the single-electron density (and energy/free energy), but not electron–electron pair correlation functions (and their corresponding Fourier transforms, which are the static structure factors). To avoid any ambiguity, we now explicitly distinguish between **pair correlation functions**, on the one hand, and the **exchange–correlation functional** on the other hand.

(7) In Fig. 5: there is no "dotted red curve" as mentioned. What do the dotted black lines represent?

We thank the referee for spotting this inconsistency. The typo "dotted red curve" has been corrected. The dotted black lines served as a guide-to-the-eye during our analysis. We now set it to the uncorrelated value of $S_{ee} \equiv 1$ as a guide-to-the-eye for the reader and added a corresponding explanation to the figure caption.

(8) On page 12: why is the Rayleigh weight "thus more suitable" for DFT-MD simulations? Please clarify the reasoning.

The Rayleigh weight $W_R(q)$ only involves $S_{eI}(q)$ and $S_{II}(q)$, i.e., pair correlations between electrons and ions, as well as between ions and ions; both of these properties are routinely estimated within DFT-MD simulations as they only require the single-electron density. In stark contrast, estimating I_{el}/I_{inel} also requires one to estimate $S_{ee}(q)$, i.e., an electron–electron pair correlation function. This is exceedingly difficult in DFT, since it is an effective single-electron theory. We have added a corresponding explanation to the ms.

(9) In Eq. 9: please explain what σ_N and S_N represent.

In Eq. (9), σ_N is a particular element from the permutation group S_N ; this explanation has been added to the ms.

(10) instead of merely claiming "our results can quantify the nodal error in the restricted PIMC approach", this should be explicitly addressed in the research. I encourage the authors to quantify and clearly demonstrate whether the node error has a significant impact, and if so, how much it matters.

As we have stated above, we do not wish to make any statements regarding actual nodal errors in the present ms. We have re-phrased the corresponding passage to "...our results can quantify potential nodal errors" to communicate to the reader the importance of a future comparison of results without any implication about the magnitude / existence of nodal errors in the regime explored in the present work.

Comments by Referee 2

The manuscript titled "Unraveling Electronic Correlations in Warm Dense Quantum Plasmas" presents significant progress in the field of plasma physics, specifically regarding the challenging theoretical description of WDM. The authors introduce a novel path integral Monte Carlo approach that bypasses nodal restrictions, enabling a more comprehensive exploration of electronic correlations in light elements under extreme conditions. Their application to strongly compressed beryllium and the comparison with experimental X-ray Thomson scattering data from the National Ignition Facility are particularly compelling. The study's findings demonstrate excellent agreement between simulation and experiment, underscoring the reliability of the method without reliance on empirical parameters. Overall, this work contributes valuable insights into electronic structures in WDM, but it is somewhat questionable to publish it in Nature Communication unless they provide strong answers to my questions below and corresponding revision on the manuscript:

We thank the referee for their positive assessment of our work, and for the suggested improvements.

A) The key point of this study is that it resolves the issue of the traditional PIMC method being underutilized in warm dense matter (WDM) studies due to challenges such as the fermion sign problem, by employing the novel approach of "carrying out a controlled extrapolation that is substituted into the canonical partition function." However, this methodology has already been presented in detail in the author's previously published references [28] and [29], and is thoroughly described in reference [35], which diminishes the originality of the paper.

We agree with the referee that the employed ξ -extrapolation method to deal with the fermion sign problem by itself is not new. However, while the method had previously been applied to considerably simpler systems (quantum harmonic oscillator and uniform electron gas, i.e., quantum one-component plasma), here we have applied it to a real system for the first time. In particular, its application to strongly compressed Be with hundreds of electrons constitutes a true mile stone and unambiguously demonstrates the value of this method for practical applications. **Moreover,**

the present work constitutes, to the best of our knowledge, the first application of PIMC (be it restricted PIMC, the ξ -extrapolation, or any other flavor) to the interpretation of XRTS experiments, which, by itself, opens up entirely new avenues away from previously used, but often questionable chemical models to robust *ab initio* simulations. Indeed, our proposed workflow of first extracting the temperature model-free from the ITCF $F(q, \tau)$ and subsequently determining the density from the ratio $I_{\text{el}}/I_{\text{inel}}$ can also be pursued with the restricted PIMC method in future works. These aspects have been emphasized more clearly in the revised ms.

B) Another aspect of novelty lies in the application of this methodology to X-ray Thomson scattering data from the National Ignition Facility (NIF). However, this too has been extensively discussed in reference [32], further reducing the originality of the work. Therefore, to emphasize the novelty of the paper, I believe it would be much more effective if the overall narrative were refocused on the new physical insights embedded in the NIF beryllium data that had not been discovered previously.

We agree with the referee. Therefore, we have carried out extensive additional PIMC calculations a) to address the inter-dependence of density and temperature for the interpretation of $I_{\text{el}}/I_{\text{inel}}$, see the revised Fig. 3, and b) to estimate the effective degree of ionization Z^* from the electrons' capacity to react to a static external perturbation in the linear response regime. The latter aspect is presented in detail in the new Fig. 4, that shows Z^* as a function of the mass density for the backscattering XRTS dataset at 120° . In particular, this study allows us to re-assess previous conclusions about the validity of common EOS models such as Steward-Pyatt and OPAL, as well as *ab initio* DFT simulations, drawn in the original paper by Döppner *et al.*

Overall, we find high consistency between our new PIMC results and DFT-MD, both of which do not inherently rely on the decomposition into effectively *bound* and *free* electrons. In contrast, these *ab initio* results substantially differ from previously used chemical models by predicting a substantially lower mass density ($\rho = 22 \pm 2 \text{ g/cc}$ vs $\rho = 34 \pm 4 \text{ g/cc}$ in the original paper), **which has very important implications for the future interpretation of XRTS measurements, and for integrated radiation hydrodynamics simulations of fusion experiments and other applications.**

C) Additionally, although this may be challenging due to the limitations of the NIF experiment, the experimental data seems insufficient to fully support the theoretical claims. In particular, in Figures 2(c) and (f) and Figure 3, there is only one experimental data point for each angle. It would be helpful to provide additional experimental data points or related experimental results to more convincingly validate the theory.

We agree with the referee that additional NIF data would be helpful; unfortunately, the utilized NIF GigaBar XRTS platform presently only supports XRTS measurements at a single scattering angle per shot.

D) Other minor comments and questions are as follows: 1. What does "XC" stand for? If it refers to Exchange Correlation, it would be helpful to clarify this for non-experts.

The abbreviation "XC" is now properly defined at the first mention of "exchange–correlation" in the main text.

2. It is impressive that a model-free temperature estimation can be made using ITCF. What is the uncertainty associated with this method for estimating the temperature (for each angle)? How many significant figures can be reliably reported? Furthermore, how does this relate to the temperature distribution observed in the experiment?

The model-free temperature estimates are given by $T = 155.5 \pm 15$ eV and $T = 190 \pm 20$ eV for the 120° and 75° datasets, respectively. These uncertainty ranges have been estimated empirically from the convergence behavior of the minimum in the ITCF with respect to the integration range x (Fig. 5 in the revised ms) and from the impact of the combined source-and-instrument function. Moreover, like in the original paper by Döppner *et al.*, we assume a single bulk temperature. This is well justified at the measurement time of $t = 0.61$ ns after peak X-ray intensity as the rebound shock has progressed through most of the material.

This is now made explicitly clear in the corresponding Methods Section.

3. What is the density of the NIF data in Figure 2?

The mass density is a-priori unknown, and determined in the now extended Figure 3 of the revised ms. We find $\rho = 22 \pm 2$ g/cc for both shots.

4. Figure 4: Why was the Chihara model chosen for comparison? Is this model regarded as the best-known standard? Although the data falls within the uncertainty range, why do the Chihara model and the author's model show greater discrepancies at small angles? Is it because small angles are sensitive to electronic correlations? If so, which specific physical factors determining the correlation function are particularly sensitive, leading to such discrepancies? How does the Chihara model treat those physical factors?

We choose to compare with the Chihara model, because it does indeed constitute, to our knowledge, the most widely used approach for the interpretation of XRTS in past studies, and because it was employed in the work by Döppner *et al.*

Note that the x -axis in the figure (Fig. 5 of the revised ms) corresponds to the integration interval for the ITCF method, cf. Eq. (4), whereas the Chihara values do not depend on x and are included as a reference. This has now been made explicitly clear in the corresponding figure caption.

5. What does "W" in Equation 8 refer to? If it is the Ewald potential, please state this explicitly. Additionally, if possible, I suggest altering the notation, as it could be confused with the Rayleigh weight.

We have re-labeled the Ewald potential as $\Phi_E(r_1, r_2)$ and now explicitly explain its meaning below Eq. (8).

Comments by Referee 3

This paper applies a recently developed method for simulation of relatively large numbers of Fermionic quantum particles. Typically this is accomplished using Kohn-Sham density functional theory (DFT), but this introduces a variety of well-known limitations including an uncontrolled approximation for the exchange-correlation functional and the Born-Oppenheimer approximations. The alternative path-integral Monte Carlo (PIMC) method avoids these limitations, but at a computational cost that scales exponentially in the number of fermions. Building on the original idea of bosonic to fermionic extrapolation (Xiong and Xiong, 2022), the authors have developed a practical implementation of the method for simulating strongly coupled electron-ion fluids at high temperature a.k.a. warm dense matter (WDM). Using careful benchmarking against exact PIMC results for small systems, they show that the extrapolation accuracy is very good and insensitive to system size. They subsequently apply the method to calculate the spatial and temporal correlations of electrons and ions in beryllium at several WDM conditions. They are able to use these results to directly compare with x-ray Thomson scattering (XRTS) data collected from shots at the National Ignition Facility

The work is highly original and the quality is very high. I anticipate that this paper will become highly cited for two reasons:

1. It establishes the validity and usefulness of an important new method for directly simulating warm dense matter
2. It provides insight in to recent experiments at NIF that are of fundamental scientific importance in astrophysics and other fields.

We thank the referee for their very positive assessment of our work.

Response to referee reports: “Unraveling electronic correlations in warm dense quantum plasmas ”

T. Dornheim *et al.*

April 9, 2025

Dear Editor,

we would like to thank both referees for reviewing our manuscript a second time, and for the very positive feedback. Please find our detailed response below.

Sincerely,

T. Dornheim, for the authors

Comments by Referee 1

The authors have significantly improved the manuscript's quality and clarity, addressing all my previous comments satisfactorily.

We thank the referee for their effort and the constructive and positive feedback during both rounds of review.

However, I still notice the following issues that need to be corrected in the revised manuscript:

1. On page 3, the example and reference in "...and experimentally verified, e.g., along the Hugoniot of Si [26]" are not accurate. A better example would be CH (Zhang et al., JCP2017), which was verified by the experiments of Döppner et al., PRL2018, and Kritcher et al., Nature 2020.

We have updated the references as suggested.

2. In the legend of Fig. 3, the theta values, instead of T, should be used to accurately label the NIF data (blue crosses) to reflect that the temperature in those experiments was not constrained to those values until this research.

We have included the values for $\Theta = k_B T / E_F$ [which depend on the mass density ρ] into Fig. 3 and added a corresponding note to the caption. In addition, we have added the "experimental" values for Θ that correspond to the estimated mass densities to the caption.

3. In Fig. 7 and the corresponding discussions, it would be better to use "eV" and "Å⁻¹" for the units of A and q, respectively, to maintain consistency with other parts of the manuscript.

We have changed the labels in Fig. 7 and adapted the corresponding discussion as requested by the referee.

4. In the caption of Fig. 7, "a free fraction of electrons" should be changed to "a fraction of free electrons."

We have changed the caption as requested.

5. The legend for PIMC data in Fig. 7 should be updated to "PIMC ($N_{Be} = 4$)" for green crosses and "PIMC ($N_{Be} = 10$)" for red circles for clarity.

We have updated the legend of Fig. 7 as requested.

6. On page 15, "as these conditions" should be changed to "at these conditions"; also, there is a redundant word in "not not respond to."

We have fixed both typos.

Comments by Referee 2

This study employs a novel ab initio Path Integral Monte Carlo (PIMC) simulation to analyze two X-ray Thomson Scattering (XRTS) datasets of strongly compressed beryllium obtained from the National Ignition Facility (NIF). The authors have adequately addressed my previous concerns regarding the novelty of their methodology, convincingly demonstrating that this is the first application of PIMC (including restricted PIMC, ξ -extrapolation, or other variants) to the interpretation of XRTS experiments. This represents a substantial advancement, as it highlights the potential of establishing robust ab initio simulations for interpreting such data. Furthermore, the additional PIMC calculations carried out by the authors have provided a novel approach to reassess the validity of commonly used EOS models, such as Steward-Pyatt and OPAL, as employed in the original work by Döppner et al. This contribution is particularly valuable as it offers an opportunity to re-evaluate existing models and experimental interpretations, paving the way for significant advancements in future studies. In conclusion, this manuscript demonstrates both academic originality and practical contributions, making it a strong candidate for publication in your journal. I wholeheartedly recommend its acceptance.

We thank the referee for both rounds of review, and for their very positive feedback.